# RLx2: Training a Sparse Deep Reinforcement Learning Model from Scratch

**Yiqin Tan**[*], **Pihe Hu**[*], **Ling Pan, Jiatai Huang, Longbo Huang**[†]
Institute for Interdisciplinary Institute for Interdisciplinary Information Sciences
Tsinghua University, Beijing, China
`{tyq22, hph19}@mails.tsinghua.edu.cn, longbohuang@tsinghua.edu.cn`

## Abstract

Training deep reinforcement learning (DRL) models usually requires high computation costs. Therefore, compressing DRL models possesses immense potential for training acceleration and model deployment. However, existing methods that generate small models mainly adopt the knowledge distillation-based approach by iteratively training a dense network. As a result, the training process still demands massive computing resources. Indeed, sparse training from scratch in DRL has not been well explored and is particularly challenging due to non-stationarity in bootstrap training. In this work, we propose a novel sparse DRL training framework, "the **R**igged **R**einforcement **L**earning **L**ottery" (RLx2), which builds upon gradient-based topology evolution and is capable of training a DRL model based entirely on sparse networks. Specifically, RLx2 introduces a novel delayed multi-step TD target mechanism with a dynamic-capacity replay buffer to achieve robust value learning and efficient topology exploration in sparse models. It also reaches state-of-the-art sparse training performance in several tasks, showing $7.5\times$-$20\times$ model compression with less than 3% performance degradation and up to $20\times$ and $50\times$ FLOPs reduction for training and inference, respectively.

## 1 Introduction

Deep reinforcement learning (DRL) has found successful applications in many important areas, e.g., games (Silver et al., 2017), robotics(Gu et al., 2017) and nuclear fusion (Degrave et al., 2022). However, training a DRL model demands heavy computational resources. For instance, AlphaGo-Zero for Go games (Silver et al., 2017), which defeats all Go-AIs and human experts, requires more than 40 days of training time on four tensor processing units (TPUs). The heavy resource requirement results in expensive consumption and hinders the application of DRL on resource-limited devices.

Sparse networks, initially proposed in deep supervised learning, have demonstrated great potential for model compression and training acceleration of deep reinforcement learning. Specifically, in deep supervised learning, the state-of-the-art sparse training frameworks, e.g., SET (Mocanu et al., 2018) and RigL (Evci et al., 2020), can train a 90%-sparse network (i.e., the resulting network size is 10% of the original network) from scratch without performance degradation. On the DRL side, existing works including Rusu et al. (2016); Schmitt et al. (2018); Zhang et al. (2019) succeeded in generating ultimately sparse DRL networks. Yet, their approaches still require iteratively training dense networks, e.g., pre-trained dense teachers may be needed. As a result, the training cost for DRL remains prohibitively high, and existing methods cannot be directly implemented on resource-limited devices, leading to low flexibility in adapting the compressed DRL models to new environments, i.e., on-device models have to be retrained at large servers and re-deployed.

Training a sparse DRL model from scratch, if done perfectly, has the potential to significantly reduce computation expenditure and enable efficient deployment on resource-limited devices, and achieves excellent flexibility in model adaptation. However, training an ultra sparse network (e.g., 90% sparsity) from scratch in DRL is challenging due to the non-stationarity in bootstrap training. Specifically, in DRL, the learning target is not fixed but evolves in a bootstrap way (Tesauro

---

[*]Equal contribution.
[†]Corresponding author.

et al., 1995), and the distribution of the training data can also be non-stationary (Desai et al., 2019). Moreover, using a sparse network structure means searching in a smaller hypothesis space, which further reduces the learning target's confidence. As a result, improper sparsification can cause irreversible damage to the learning path (Igl et al., 2021), resulting in poor performance. Indeed, recent works (Sokar et al., 2021; Graesser et al., 2022) show that a direct adoption of a dynamic sparse training (DST) framework in DRL still fails to achieve good compression of the model for different environments uniformly. Therefore, the following interesting question remains open:

*Can an efficient DRL agent be trained from scratch with an ultra-sparse network throughout?*

In this paper, we give an affirmative answer to the problem and propose a novel sparse training framework, "the **R**igged **R**einforcement **L**earning **L**ottery" (RLx2), for off-policy RL, which is the first algorithm to achieve sparse training throughout using sparsity of more than 90% with only minimal performance loss. RLx2 is inspired by the gradient-based topology evolution criteria in RigL (Evci et al., 2020) for supervised learning. However, a direct application of RigL does not achieve high sparsity, because sparse DRL models suffer from unreliable value estimation due to limited hypothesis space, which further disturbs topology evolution. Thus, RLx2 is equipped with a delayed multi-step Temporal Difference (TD) target mechanism and a novel dynamic-capacity replay buffer to achieve robust value learning and efficient topology exploration. These two new components address the value estimation problem under sparse topology, and together with RigL, achieve superior sparse-training performance.

The main contributions of the paper are summarized as follows.

- We investigate the fundamental obstacles in training a sparse DRL agent from scratch, and discover two key factors for achieving good performance under sparse networks, namely robust value estimation and efficient topology exploration.

- Motivated by our findings, we propose RLx2, the first framework that enables DRL training based entirely on sparse networks. RLx2 possesses two key functions, i.e., a gradient-based search scheme for efficient topology exploration, and a delayed multi-step TD target mechanism with a dynamic-capacity replay buffer for robust value learning.

- Through extensive experiments, we demonstrate the state-of-the-art sparse training performance of RLx2 with two popular DRL algorithms, TD3 (Fujimoto et al., 2018) and SAC (Haarnoja et al., 2018), on several MuJoCo (Todorov et al., 2012) continuous control tasks. Our results show up to $20\times$ model compression. RLx2 also achieves $20\times$ acceleration in training and $50\times$ in inference in terms of floating-point operations (FLOPs).

## 2 RELATED WORKS

We discuss the related works on training sparse models in deep supervised learning and reinforcement learning below. We also provide a comprehensive performance comparison in Table 1.

**Sparse Models in Deep Supervised Learning**  Han et al. (2015; 2016); Srinivas et al. (2017); Zhu & Gupta (2018) focus on finding a sparse network by pruning pre-trained dense networks. Iterative Magnitude Pruning (IMP) in Han et al. (2016) achieves a sparsity of more than 90%. Techniques including neuron characteristic (Hu et al., 2016), dynamic network surgery (Guo et al., 2016), derivatives (Dong et al., 2017; Molchanov et al., 2019b), regularization (Louizos et al., 2018; Tartaglione et al., 2018), dropout (Molchanov et al., 2017), and weight reparameterizationSchwarz et al. (2021) have also been applied in network pruning. Another line of work focuses on the Lottery Ticket Hypothesis (LTH), first proposed in Frankle & Carbin (2019), which shows that training from a sparse network from scratch is possible if one finds a sparse "winning ticket" initialization in deep supervised learning. The LTH is also validated in other deep learning models (Chen et al., 2020; Brix et al., 2020; Chen et al., 2021).

Many works (Bellec et al., 2017; Mocanu et al., 2018; Mostafa & Wang, 2019; Dettmers & Zettlemoyer, 2019; Evci et al., 2020) also try to train a sparse neural network from scratch without having to pre-trained dense models. These works adjust structures of sparse networks during training, including Deep Rewiring (DeepR) (Bellec et al., 2017), Sparse Evolutionary Training (SET) (Mocanu et al., 2018), Dynamic Sparse Reparameterization (DSR) (Mostafa & Wang, 2019), Sparse

Networks from Scratch (SNFS) (Dettmers & Zettlemoyer, 2019), and Rigged Lottery (RigL) (Evci et al., 2020). Single-Shot Network Pruning (SNIP) (Lee et al., 2019) and Gradient Signal Preservation (GraSP) (Wang et al., 2020) focus on finding static sparse networks before training.

Table 1: Comparison of different sparse training techniques in DRL. Here ST and TA stand for "sparse throughout training" and "training acceleration", respectively. The shown sparsity is the maximum sparsity level without performance degradation under the algorithms. †: There are multiple method combinations in (Graesser et al., 2022), where "TE" stands for two topology evolution schemes: SET and RigL, "RL" refers to two RL algorithms: TD3 and SAC.

| Name | Paradigm | Scenario | ST | TA | Sparsity |
|---|---|---|---|---|---|
| PoPS (Livne & Cohen, 2020) | IMP | Online | No | No | $\sim 99\%$ |
| LTH-RL (Yu et al., 2020) | IMP | Online | Yes | No | $\sim 99\%$ |
| LTH-IL (Vischer et al., 2022) | IMP | Online | Yes | No | $\sim 95\%$ |
| SSP (Arnob et al., 2021) | Single-shot pruning | Offline | Yes | Yes | $\sim 95\%$ |
| GST (Lee et al., 2021) | Gradual pruning | Online | No | No | $\sim 70\%$ |
| DST (Sokar et al., 2021) | Topology Evolution | Online | Yes | Yes | $\sim 50\%$ |
| TE-RL$^\dagger$ (Graesser et al., 2022) | Topology Evolution | Online | Yes | Yes | $\sim 90\%$ |
| RLx2 (Ours) | Topology Evolution | Online | Yes | Yes | $\sim \mathbf{95\%}$ |

**Sparse Models in DRL**  Evci et al. (2020); Sokar et al. (2021) show that finding a sparse model in DRL is difficult due to training instability. Existing works (Rusu et al., 2016; Schmitt et al., 2018; Zhang et al., 2019) leverage knowledge distillation with static data to avoid unstable training and obtain small dense agents. Policy Pruning and Shrinking (PoPs) in Livne & Cohen (2020) obtains a sparse DRL agent with iterative policy pruning (similar to IMP). LTH in DRL is firstly investigated in Yu et al. (2020), and then Vischer et al. (2022) shows that a sparse winning ticket can also be found by behavior cloning (BC). Another line of works (Lee et al., 2021; Sokar et al., 2021; Arnob et al., 2021) attempts to train a sparse neural network from scratch without pre-training a dense teacher. Group Sparse Training (GST) in Lee et al. (2021) utilizes block-circuit compression and pruning. Sokar et al. (2021) proposes using SET in topology evolution in DRL and achieves 50% sparsity. Arnob et al. (2021) proposes single-shot pruning (SSP) for offline RL. Graesser et al. (2022) finds that pruning often obtains the best results and plain dynamic sparse training methods, including SET and RigL, improves over static sparse training significantly. However, existing works either demands massive computing resources, e.g. pruning-based methods (Rusu et al., 2016; Schmitt et al., 2018; Zhang et al., 2019; Livne & Cohen, 2020), or fail in ultra sparse models, e.g. DST-based methods (Sokar et al., 2021; Graesser et al., 2022). In this paper, we further improve the performance of DST by introducing a delayed multi-step TD target mechanism with a dynamic-capacity replay buffer, which effectively addresses the unreliability of fixed-topology models during sparse training.

## 3 DEEP REINFORCEMENT LEARNING PRELIMINARIES

In reinforcement learning, an agent interacts with an unknown environment to learn an optimal policy. The learning process is formulated as a Markov decision process (MDP) $\mathcal{M} = \langle \mathcal{S}, \mathcal{A}, r, \mathbb{P}, \gamma \rangle$, where $\mathcal{S}$ is the state space, $\mathcal{A}$ is the action space, $r$ is the reward function, $\mathbb{P}$ denotes the transition matrix, and $\gamma$ stands for the discount factor. Specifically, at time slot $t$, given the current state $s_t \in \mathcal{S}$, the agent selects an action $a_t \in \mathcal{A}$ by policy $\pi : \mathcal{S} \to \mathcal{A}$, which then incurs a reward $r(s_t, a_t)$.

Denote the Q function associated with the policy $\pi$ for state-action pair $(s, a)$ as

$$Q_\pi(s, a) = \mathbb{E}_\pi \left[ \sum_{i=t}^{T} \gamma^{i-t} r(s_i, a_i) | s_t = s, a_t = a \right]. \tag{1}$$

In actor-critic methods (Silver et al., 2014), the policy $\pi(s; \phi)$ is parameterized by a policy (actor) network with weight parameter $\phi$, and the Q function $Q(s, a; \theta)$ is parameterized by a value (critic) network with parameter $\theta$. The goal of the agent is to find an optimal policy $\pi^*(s; \phi^*)$ which maximizes long-term cumulative reward, i.e., $J^* = \max_\phi \mathbb{E}_{\pi(\phi)}[\sum_{i=0}^{T} \gamma^{i-t} r(s_i, a_i) | s_0, a_0]$.

There are various DRL methods for learning an efficient policy. In this paper, we focus on off-policy TD learning methods, including a broad range of state-of-the-art algorithms, e.g., TD3 (Fujimoto

et al., 2018) and SAC (Haarnoja et al., 2018). Specifically, the critic network is updated by gradient descent to fit the one-step TD targets $\mathcal{T}_1$ generated by a target network $Q(s, a; \theta')$, i.e.,

$$\mathcal{T}_1 = r(s, a) + \gamma Q\left(s', a'; \theta'\right) \tag{2}$$

for each state-action pair $(s, a)$, where $a' = \pi(s'; \phi)$. The loss function of the value network is defined as the expected squared loss between the current value network and TD targets:

$$\mathcal{L}(\theta) = \mathbb{E}_{\pi(\phi)}\left[Q(s, a; \theta) - \mathcal{T}_1\right]^2. \tag{3}$$

The policy $\pi(s; \phi)$ is updated by the deterministic policy gradient algorithm in Silver et al. (2014):

$$\nabla_\phi J(\phi) = \mathbb{E}_{\pi(\phi)}\left[\left.\nabla_a Q_\pi(s, a; \theta)\right|_{a=\pi(s;\phi)} \nabla_\phi \pi(s; \phi)\right].$$

## 4 RLx2: RIGGING THE LOTTERY IN DRL

In this section, we present the RLx2 algorithm, which is capable of training a sparse DRL model from scratch. An overview of the RLx2 framework on an actor-critic architecture is shown in Figure 1. To motivate the design of RLx2, we present a comparison of four sparse DRL training methods using TD3 with different topology update schemes on InvertedPendulum-v2, a simple control task from MuJoCo, in Figure 2.[1]

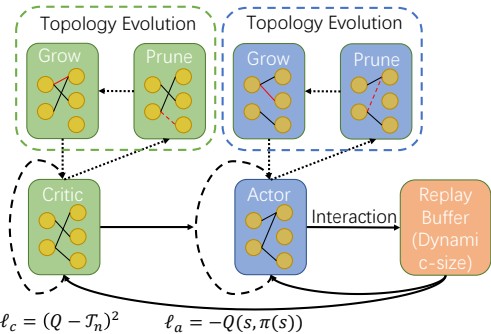

Figure 1: The RLx2 framework contains three key components, i.e., multi-step TD target mechanism, dynamic-capacity replay buffer and gradient-based topology evolution.

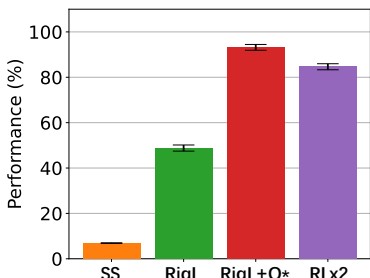

Figure 2: Performance comparison for four sparse training methods, i.e., SS, RigL, RigL+$Q^*$ and RLx2. The results show that both efficient topology evolution and robust value estimation are critical.

From the results, we make the following important observations. (i) *Topology evolution is essential*. It can be seen that a random static sparse network (SS) leads to much worse performance than RigL. (ii) *Robust value estimation is significant*. This is validated by the comparison between RigL and RigL+$Q^*$, both using the same topology adjustment scheme but with different Q-values.

Motivated by the above findings, RLx2 utilizes gradient-based topology adjustment, i.e., RigL (for topology evolution), and introduces a delayed multi-step TD target mechanism with a dynamic-capacity replay buffer (for robust value estimation). Below, we explain the key components of RLx2 in detail, to illustrate why RLx2 is capable of achieving robust value learning and efficient topology exploration simultaneously.

### 4.1 GRADIENT-BASED TOPOLOGY EVOLUTION

The topology evolution in RLx2 is conducted by adopting the RigL method (Evci et al., 2020). Specifically, we compute the gradient values of the loss function with respect to link weights. Then, we dynamically grow connections (connecting neurons) with large gradients and remove existing links with the smallest absolute value of the weights. In this way, we obtain a sparse mask that evolves by self-adjustment.

---

[1]Four schemes: 1) training with a random static sparse network (SS); 2) training with RigL, (RigL); 3) dynamic sparse training guided by true Q-value, i.e., Q-values from a fully trained expert critic with a dense network (RigL+$Q^*$); 4) and dynamic sparse training guided by learned Q-value with TD targets (RLx2).

The pseudo-code of our scheme is given in Algorithm 1, where $\odot$ is the element-wise multiplication operator and $M_\theta$ is the binary mask to represent the sparse topology of the network $\theta$. The update fraction anneals during the training process according to $\zeta_t = \frac{\zeta_0}{2}(1 + \cos(\frac{\pi t}{T_{\text{end}}}))$, where $\zeta_0$ is the initial update fraction and $T_{\text{end}}$ is the total number of iterations. Finding top-$k$ links with maximum gradients in Line 10 can be efficiently implemented such that Algorithm 1 owns time complexity $O((1-s)N \log N))$ (detailed in Appendix A.1), where $s$ is the total sparsity. Besides, the topology adjustment happens very infrequently during the training, i.e., every 10000 step in our setup, such that consumption of this step is negligible (detailed in Appendix C.3). Our topology evolution scheme can be implemented efficiently on resource-limited devices.

---

**Algorithm 1** Topology Evolution (Evci et al., 2020)

1:  $N_l$: Number of parameters in layer $l$
2:  $\theta_l$: Parameters in layer $l$
3:  $M_{\theta_l}$: Sparse mask of layer $l$
4:  $s_l$: Sparsity of layer $l$
5:  $L$: Loss function
6:  $\zeta_t$: Update fraction in training step $t$
7:  **for** each layer $l$ **do**
8:      $k = \zeta_t(1 - s_l)N_l$
9:      $\mathbb{I}_{\text{drop}} = \text{ArgTopK}(-|\theta_l \odot M_{\theta_l}|, k)$
10:     $\mathbb{I}_{\text{grow}} = \text{ArgTopK}_{i \notin \theta_l \odot M_{\theta_l} \setminus \mathbb{I}_{\text{drop}}}(|\nabla_{\theta_l}L, k|)$
11:    Update $M_{\theta_l}$ according to $\mathbb{I}_{\text{drop}}$ and $\mathbb{I}_{\text{grow}}$
12:     $\theta_l \leftarrow \theta_l \odot M_{\theta_l}$
13: **end for**

---

### 4.2  ROBUST VALUE LEARNING

As discussed above, value function learning is crucial in sparse training. Specifically, we find that under sparse models, robust value learning not only serves to guarantee the efficiency of bootstrap training as in dense models, but also guides the gradient-based topology exploration of the sparse network during training.

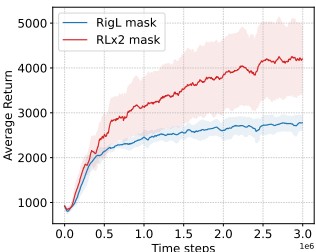

Figure 3: Sparse model comparison in Ant-v3.

Figure 3 compares the performance of the masks (i.e., sparse network topology) obtained by RigL and RLx2 (i.e., RigL + **r**obust value **l**earning) on Ant-v3. Here we use a method similar to (Frankle & Carbin, 2019) for evaluating the obtained sparse mask:[2] 1) first initialize a random sparse topology; 2) keep adjusting the topology during the training and obtain the final mask; 3) train a sparse agent with the obtained mask (the mask is fixed throughout this training phase, only the weights are restored to their initial values as in the first step at the beginning). It can be clearly observed that the mask by RLx2 significantly outperforms that by solely using RigL (Appendix C.4 provides details and experiments in other environments, where similar results are observed).

To achieve robust value estimation and properly guide the topology search, RLx2 utilizes two novel components: i) delayed multi-step TD targets to bootstrap value estimation; ii) a dynamic-capacity replay buffer to eliminate the potential data inconsistency due to policy change during training.

#### 4.2.1  MULTI-STEP TD TARGET

In TD learning, a TD target is generated, and the value network will be iteratively updated by minimizing a squared loss induced by the TD target. Single-step methods generate the TD target by combining one-step reward and discounted target network output, i.e., $\mathcal{T}_1 = r_t + \gamma Q(s_{t+1}, \pi(s_{t+1}); \theta)$. However, a sparse network parameter $\widehat{\theta} = \theta \odot M_\theta$, obtained from its dense counterpart $\theta$, will inevitably reside in a smaller hypothesis space due to using fewer parameters. This means that the output of the sparse value network $\widehat{\theta}$ can be unreliable and may lead to inaccurate value estimation. Denote the fitting error of the value network as $\epsilon(s, a) = Q(s, a; \theta) - Q_\pi(s, a)$. One sees that this error may be larger under a sparse model compared to that under a dense network.

To overcome this issue, we adopt a multi-step target, i.e., $\mathcal{T}_n = \sum_{k=0}^{n-1} \gamma^k r_{t+k} + \gamma^n Q(s_{t+n}, \pi(s_{t+n}); \theta)$, where the target combines an $N$-step sample and the output of the sparse value network after $N$-step, both appropriately discounted. By doing so, we reduce the expected error between the TD target and the true target. Specifically, Eq.(4) shows the expected TD error between multi-step TD target $\mathcal{T}_n$ and the true Q-value $Q_\pi$ associated with the target policy $\pi$, conditioned on transitions from behavior policy $b$ (see detailed derivation in Appendix A.2).

$$\mathbb{E}_b[\mathcal{T}_n(s,a)] - Q_\pi(s,a) = \underbrace{(\mathbb{E}_b[\mathcal{T}_n(s,a)] - \mathbb{E}_\pi[\mathcal{T}_n(s,a)])}_{\text{Policy inconsistency error}} + \gamma^n \underbrace{\mathbb{E}_\pi[\epsilon(s_n, \pi(s_n))]}_{\text{Network fitting error}} \quad (4)$$

---

[2](Frankle & Carbin, 2019) obtains the mask, i.e., the "lottery ticket", by pruning a pretrained dense model. Our sparse mask is the final mask obtained by dynamic sparse training.

The multi-step target has been studied in existing works (Bertsekas & Ioffe, 1996; Precup, 2000; Munos et al., 2016) for improving TD learning. In our case, we also find that introducing a multi-step target reduces the network fitting error by a multiplicative factor $\gamma^n$, as shown in Eq. (4). On the other hand, it has been observed, e.g., in Fedus et al. (2020), that an immediate adoption of multi-step TD targets may cause a larger policy inconsistency error (the first term in Eq. (4)). Thus, we adopt a delayed scheme to suppress policy inconsistency and further improve value learning. Specifically, at the early stage of training, we use one-step TD targets to better handle the quickly changing policy during this period, where a multi-step target may not be meaningful. Then, after several training epochs, when the policy change becomes less abrupt, We permanently switch to multi-step TD targets, to exploit its better approximation of the value function.

### 4.2.2 DYNAMIC-CAPACITY BUFFER

The second component of RLx2 for robust value learning is a novel dynamic buffer scheme for controlling data inconsistency. Off-policy algorithms use a replay buffer to store collected data and train networks with sampled batches from the buffer. Their performances generally improve when larger replay capacities are used (Fedus et al., 2020). However, off-policy algorithms with unlimited-size replay buffers can suffer from policy inconsistency due to the following two aspects.

(i) *Inconsistent multi-step targets:* In off-policy algorithms with multi-step TD targets, the value function is updated to minimize the squared loss in Eq. (3) on transitions sampled from the replay buffer, i.e., the reward sequence $r_t, r_{t+1}, \cdots, r_{t+n}$ collected during training. However, the fact that the policy can evolve during training means that the data in the replay buffer, used for Monte-Carlo approximation of the current policy $\pi$, may be collected under a different behavior policy $b$ (Hernandez-Garcia & Sutton, 2019; Fedus et al., 2020). As a result, it may lead to a large policy inconsistency error in Eq. (4), causing inaccurate estimation.

(ii) *Mismatched training data:* In practice, the agent minimizes the value loss $\widehat{\mathcal{L}}(\theta)$ with respect to the sampled value in mini-batch $\mathcal{B}_t$, given by

$$\widehat{\mathcal{L}}(\theta) = \frac{1}{|\mathcal{B}_t|} \sum_{(s_i, a_i) \sim \mathcal{B}_t} (Q(s_i, a_i; \theta) - \mathcal{T})^2 \qquad (5)$$

Compared to Eq. (3), the difference between the distribution of transitions in the mini-batch $\mathcal{B}_t$ and the true transition distribution induced by the current policy also leads to a mismatch in the training objective (Fujimoto et al., 2019). Indeed, our analysis in Appendix A.4 shows that training performance is closely connected to policy consistency.

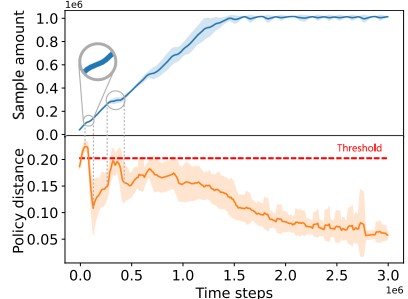

Figure 4: Dynamic buffer capacity & policy inconsistency

Motivated by our analysis, we introduce a dynamically-sized buffer to reduce the policy gap based on the policy distance of the collected data. The formal scheme is given in Algorithm 3. We introduce the following *policy distance measure* to evaluate the inconsistency of data in the buffer, i.e.,

$$\mathcal{D}(\mathcal{B}, \phi) = \frac{1}{K} \sum_{(s_i, a_i) \in \text{OldK}(\mathcal{B})} \| \pi(s_i; \phi) - a_i \|_2, \qquad (6)$$

where $\mathcal{B}$ denotes the current replay buffer, $\text{OldK}(\mathcal{B})$ denotes the oldest $K$ transitions in $\mathcal{B}$, and $\pi(\cdot; \phi)$ is the current policy. Here $K$ is a hyperparameter. We calculate the $\mathcal{D}(\mathcal{B}, \phi)$ value every $\Delta_b$ steps. If $\mathcal{D}(\mathcal{B}, \phi)$ gets above a certain pre-specified threshold $D_0$, we start to pop items from $\mathcal{B}$ in a First-In-First-Out (FIFO) order until this distance measure $\mathcal{D}$ becomes below the threshold.

A visualization of the number of stored samples and the proposed policy distance metric during training is shown in Figure 4. We see that the policy distance oscillates in the early stage as the policy evolves, but it is tightly controlled and does not violate the threshold condition to effectively address the off-policyness issue. As the policy converges, the policy distance tends to decrease and converge (Appendix C.8 also shows that the performance is insensitive to the policy threshold $D_0$).

## 5 EXPERIMENTS

In this section, we investigate the performance improvement of RLx2 in Section 5.1, and the importance of each component in RLx2 in Section 5.2. In particular, we pay extra attention to the role

topology evolution plays in sparse training in Section 5.3. Our experiments are conducted in four popular MuJoCo environments: HalfCheetah-v3 (Hal.), Hopper-v3 (Hop.), Walker2d-v3 (Wal.), and Ant-v3 (Ant.),[3] for RLx2 with two off-policy algorithms, TD3 and SAC. Instantiations of RLx2 on TD3 and SAC are provided in Appendix B. Each result is averaged over eight random seeds. The code is available at `https://github.com/tyq1024/RLx2`.

## 5.1 COMPARATIVE EVALUATION

Table 2 summarizes the comparison results. In our experiments, we compare RLx2 with the following baselines: (i) **Tiny**, which uses tiny dense networks with the same number of parameters as the sparse model in training. (ii) **SS**: using static sparse networks with random initialization. (iii) **SET** (Mocanu et al., 2018), which uses dynamic sparse training by dropping connections according to the magnitude and growing connections randomly. Please notice that the previous work (Sokar et al., 2021) also adopts the SET algorithm for topology evolution in reinforcement learning. Our implementations reach better performance due to different hyperparameters. (iv) **RigL** (Evci et al., 2020), which uses dynamic sparse training by dropping and growing connections with magnitude and gradient criteria, respectively, the same as RLx2's topology evolution procedure.

Table 2: Comparisons of RLx2 with sparse training baselines. Here "Sp." refers to the sparsity level (percentage of model size reduced), "Total Size" refers to the total parameters of both critic and actor networks (detailed calculation of training and inference FLOPs are given in Appendix C.3). The right five columns show the final performance of different methods. The "Total size," "FLOPs" , and "Performance" are all normalized w.r.t. the original large dense model (detailed in Appendix C.2).

| Alg. | Env. | Actor Sp. | Critic Sp. | Total Size | FLOPs (Train) | FLOPs (Test) | Tiny (%) | SS (%) | SET (%) | RigL (%) | RLx2 (%) |
|------|------|-----------|------------|------------|---------------|--------------|----------|--------|---------|----------|----------|
| TD3 | Hal. | 90% | 85% | 0.133x | 0.138x | 0.100x | 86.3 | 77.1 | 92.6 | 90.8 | **99.8** |
|  | Hop. | 98% | 95% | 0.040x | 0.043x | 0.020x | 64.5 | 67.7 | 66.5 | 90.6 | **97.0** |
|  | Wal. | 97% | 95% | 0.043x | 0.045x | 0.030x | 60.8 | 42.9 | 39.3 | 35.7 | **98.1** |
|  | Ant. | 96% | 88% | 0.093x | 0.100x | 0.040x | 16.5 | 49.6 | 62.5 | 68.5 | **103.9** |
|  | Avg. | 95% | 91% | 0.077x | 0.081x | 0.048x | 57.0 | 59.3 | 65.2 | 71.4 | **99.7** |
| SAC | Hal. | 90% | 80% | 0.180x | 0.197x | 0.100x | 95.0 | 75.4 | 94.8 | 89.8 | **102.2** |
|  | Hop. | 98% | 95% | 0.044x | 0.048x | 0.020x | 89.1 | 81.6 | 103.9 | **110.0** | 109.7 |
|  | Wal. | 90% | 90% | 0.100x | 0.113x | 0.100x | 73.8 | 83.4 | 95.8 | 81.9 | **103.2** |
|  | Ant | 90% | 75% | 0.220x | 0.239x | 0.100x | 49.6 | 49.3 | 79.8 | 90.9 | **105.6** |
|  | Avg. | 92% | 85% | 0.136x | 0.149x | 0.080x | 76.9 | 72.4 | 93.6 | 93.2 | **105.2** |
| Avg. |  | 94% | 88% | 0.107x | 0.115x | 0.064x | 67.0 | 65.9 | 79.4 | 82.3 | **101.8** |

In our experiments, we allow the actor and critic networks to take different sparsities. We define an ultimate compression ratio, i.e., the largest sparsity level under which the performance degradation under RLx2 is within $\pm\%3$ of that under the original dense models. This can also be understood as the minimum size of the sparse model with the full performance of the original dense model. We present performance comparison results in Table 2 based on the ultimate compression ratio. The performance of each algorithm is evaluated with the average reward per episode over the last 30 policy evaluations of the training ( policy evaluation is conducted every 5000 steps). Hyperparameters are fixed in all four environments for TD3 and SAC, respectively, which are presented in Appendix C.2.

**Performance**    Table 2 shows RLx2 performs best among all baselines in all four environments by a large margin (except for a close performance with RigL with SAC in Hopper). In addition, tiny dense (Tiny) and random static sparse networks (SS) performance are worst on average. SET and RigL are better yet fail to maintain the performance in Walker2d-v3 and Ant-v3, which means robust value learning is necessary under sparse training. To further validate the performance of RLx2, we compare the performance of different methods under different sparsity levels in Hopper-v3 and Ant-v3 in Figure 5, showing RLx2 has a significant performance gain over other baselines.

**Model Compression**    RLx2 achieves superior compression ratios (the reciprocal of the total size), with minor performance degradation (less than 3%). Specifically, RLx2 with TD3 achieves 7.5×- 25× model compression, with the best compression ratios of 25× on Hopper-v3. The actor can be

---

[3]A more complex environment with larger state space, Humanoid-v3, is also evaluated in Appendix C.9.

compressed for each environment by more than $96\%$, and the critic is compressed by $85\%$-$95\%$. The results for SAC are similar. RLx2 with SAC achieves a $5\times$-$20\times$ model compression.

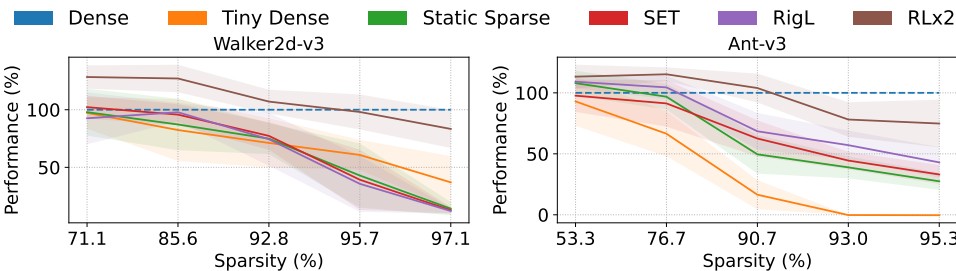

Figure 5: Performance comparison under different model sparsity.

**Acceleration in FLOPs**   Different from knowledge-distillation/BC based methods, e.g., Livne & Cohen (2020); Vischer et al. (2022); Lee et al. (2021), RLx2 uses a sparse network *throughout* training. Thus, it has an additional advantage of immensely accelerating training and saving computation, i.e., $12\times$ training acceleration and $20\times$ inference acceleration for RLx2-TD3, and $7\times$ training acceleration and $12\times$ inference acceleration for RLx2-SAC.

## 5.2 ABLATION STUDY

We conduct a comprehensive ablation study on the three critical components of RLx2 on TD3, i.e., topology evolution, multi-step TD target, and dynamic-capacity buffer, to examine the effect of each component in RLx2 and their robustness in hyperparameters. In addition, we provide the sensitivity analysis for algorithm hyper-parameters, e.g. initial mask update fraction, mask update interval, buffer adjustment interval, and buffer policy distance threshold, in Appendix C.8.

**Topology evolution**   RLx2 drops and grows connections with magnitude and gradient criteria, respectively, which has been adopted in RigL (Evci et al., 2020) for deep supervised learning. To validate the necessity of our topology evolution criteria, we compare RLx2 with three baselines, which replace the topology evolution scheme in RLx2 with Tiny, SS and SET, while keeping other components in RLx2 unchanged.[4] Thy evolution e left partpolog of Table 3 shows that RigL as a topology adjustment scheme (the resulting scheme is RLx2 when using RigL) performs best among the four baselines. We also observe that Tiny performs worst, which is consistent with the conclusion in existing works (Zhu & Gupta, 2018) that a sparse network may contain a smaller hypothesis space and leads to performance loss, which necessitates a topology evolution scheme.

Table 3: Ablation study on topology evolution and multi-step target, where the performance (%) is normalized with respect to the performance of dense models.

| Env. | Topoloy Evolution | | | | Multi-step Target | | | | |
|------|------|------|------|------|--------|--------|--------|--------|--------|
|      | Tiny | SS | SET | **RLx2** | 1-step | 2-step | **3-step** | 4-step | 5-step |
| Hal. | 93.3 | 86.1 | 100.1 | 99.8 | 96.5 | 101.7 | 99.8 | 98.8 | 97.0 |
| Hop. | 74.4 | 84.2 | 88.8 | 97.0 | 77.9 | 91.7 | 97.0 | 84.0 | 87.5 |
| Wal. | 84.1 | 83.8 | 89.4 | 98.1 | 73.9 | 93.7 | 98.1 | 99.1 | 99.3 |
| Ant. | 28.7 | 80.2 | 83.5 | 103.9 | 103.9 | 105.1 | 103.9 | 96.7 | 94.5 |
| Avg. | 70.1 | 83.6 | 90.4 | **99.7** | 88.1 | 98.1 | **99.7** | 94.6 | 94.6 |

**Multi-step TD targets**   We also compare different step lengths in multi-step TD targets for RLx2 in the right part of Table 3. We find that multi-step TD targets with a step length of 3 obtain the maximum performance. In particular, multi-step TD targets improve the performance dramatically in Hopper-v3 and Walker2d-v3, while the improvement in HalfCheetach-v3 and Ant-v3 is minor.

**Dynamic-capacity Buffer**   We compare different buffer sizing schemes, including our dynamic scheme, different fixed-capacity buffers, and an unlimited buffer. Figure 6 shows that our dynamic-capacity buffer performs best among all settings of the buffer. Smaller

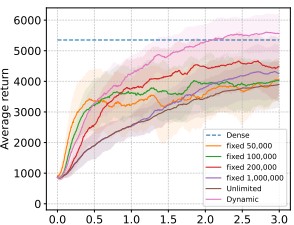

Figure 6: Performance with different buffer schemes.

---

[4]Take Algorithm 4 in Appendix B for an example. Only lines 16 and 21 of "Topology Evolution(·)" are changed while other parts remain unchanged. We also regard static topology as a special topology evolution.

buffer capacity benefits the performance in the early stage but may reduce the final performance. This is because using a smaller buffer results in higher sample efficiency in the early stage of training but fails in reaching high performance in the long term, whereas a large or even unlimited one may perform poorly in all stages.

### 5.3 WHY EVOLVE TOPOLOGY IN DRL?

Compared to dense networks, sparse networks have smaller hypothesis spaces. Even under the same sparsity, different sparse architectures correspond to different hypothesis spaces. As Frankle & Carbin (2019) has shown, some sparse architecture (e.g., the "winning ticket") performs better than a random one. To emphasize the necessity of topology evolution in sparse training, we compare different sparse network architectures in Figure 7, including the random ticket (topology sampled at random and fixed throughout training), the winning ticket (topology from an RLx2 run and fixed throughout training), and a dynamic ticket (i.e., training using RLx2) under both reinforcement learning (RL) and behavior cloning (BC).[5] From Figure 7(a), we see that RLx2 achieves the best

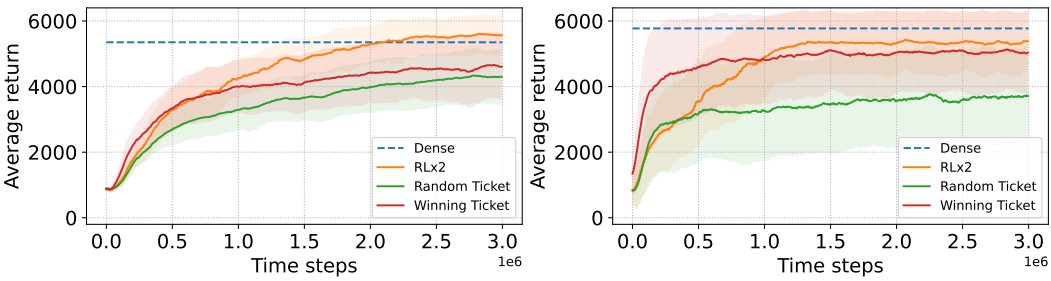

(a) Rigging the Lottery Ticket in RL training    (b) Rigging the Lottery Ticket in behavior cloning

Figure 7: Comparison of different sparse network architecture for training a sparse DRL agent in Ant-v3, where the sparsity is the same as that in Table 2.

performance, which is comparable with that under the original dense model. Due to the potential data inconsistency problem in value learning and the smaller hypothesis search space under sparse networks, training with a single fixed topology does not fully reap the benefit of high sparsity and can cause significantly degraded performance. That is why the winning ticket and random ticket both lead to significant performance loss compared to RLx2. On the other hand, Figure 7(b) shows that in BC tasks, the winning ticket and RLx2 perform almost the same as the dense model, while the random ticket performs worst. This indicates that an appropriate fixed topology can indeed be sufficient to reach satisfactory performance in BC, which is intuitive since BC adopts a supervised learning approach and eliminates non-stationarity due to bootstrapping training. In conclusion, we find that a fixed winning ticket can perform as well as a dynamic topology that evolves during the training in behavior cloning, while RLx2 outperforms the winning ticket in RL training. This observation indicates that topology evolution not only helps find the winning ticket in sparse DRL training but is also necessary for training a sparse DRL agent due to the extra non-stationary in bootstrapping training, compared to deep supervised learning.

## 6 CONCLUSION

This paper proposes a sparse training framework, RLx2, for off-policy reinforcement learning (RL). RLx2 utilizes gradient-based evolution to enable efficient topology exploration and establishes robust value learning using a delayed multi-step TD target mechanism with a dynamic-capacity replay buffer. RLx2 enables training an efficient DRL agent with minimal performance loss using an ultra-sparse network throughout training and removes the need for pre-training dense networks. Our extensive experiments on RLx2 with TD3 and SAC demonstrate state-of-the-art sparse training performance, showing a $7.5\times$-$20\times$ model compression with less than $3\%$ performance degradation and up to $20\times$ and $50\times$ FLOPs reduction in training and inference, respectively. It will be interesting future work to extend the RLx2 framework in more complex RL scenarios requiring more intense demand for computing resources, e.g., real-world problems instead of standard MuJoCo environments or multi-agent settings.

---

[5]In BC, the actor network is trained under the guidance of a well-trained expert instead of the critic network.

## REPRODUCIBILITY STATEMENT

Experiment details (including an efficient implementation for RLx2, implementation details of the dynamic buffer, hyperparameters, and network architectures) are included in Appendix C for reproduction. The proof for our analysis of the dynamic buffer can be found in Appendix A.4. The code is open-sourced in `https://github.com/tyq1024/RLx2`.

## ACKNOWLEDGEMENTS

The work is supported by the Technology and Innovation Major Project of the Ministry of Science and Technology of China under Grant 2020AAA0108400 and 2020AAA0108403, the Tsinghua University Initiative Scientific Research Program, and Tsinghua Precision Medicine Foundation 10001020109.

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

# Supplementary Materials

# A    ADDITIONAL DETAILS FOR SECTION 4

This section provides additional details for Section 4, including how to efficiently implement Algorithm 1 with limited resource, the derivation of Eq. (4) in Section 4.2.1, and the full algorithm of dynamic-capacity buffer in Section 4.2.2.

## A.1    EFFICIENT IMPLEMENTATION FOR ALGORITHM 1

For simplicity, in this section, we omit all indices of $l$ in the symbols that appeared in Algorithm 1.

**Parameter Storing**    Suppose layer $l$ takes an $n^{(in)}$-dimensional vector $\mathbf{x}$ as the input, and outputs an $n^{(out)}$-feature vector $\mathbf{y}$ via a linear transformation. Then the layer's number of parameters is $N = n^{(in)} \times n^{(out)}$. A naive implementation will be to store both $\theta \in \mathbb{R}^{n^{(out)} \times n^{(in)}}$ and $M_\theta \in \{0,1\}^{n^{(out)} \times n^{(in)}}$ in two dense $n^{(out)} \times n^{(in)}$ matrices in the memory. In forward and backward propagations, one simply performs the dense-matrix-multiply-vector operation on $\theta$ and $\mathbf{x}$. However, this implementation cannot enjoy any speed-up even when we are using a sparsity ratio $s$ close to 1, so the network is highly sparse. Also, the actual memory occupied by the model is always proportional to $N$ and irrelevant to $s$. However, a better way is to store $\theta$ in a more compact manner, where only the non-zero indices (i.e., positions of the ones in $M_\theta$) and their values. As a result, the weights of the layer now occupies $\Theta((1-s)N)$ memory, and the matrix-multiply-vector operation also just costs $O((1-s)N + n^{(in)} + n^{(out)})$. Such sparse matrix (or tensor of higher orders) structures are supported by many modern machine learning frameworks, e.g., `torch.sparse` in PyTorch. Many of them also support automatic gradient calculation and backward propagation.

**Link Dropping**    With this sparse representation of $\theta$, it can be seen that the link dropping step of Algorithm 1 (Line 9) can be done in $O((1-s)N \log N)$ time by sorting all the weight entries in their absolute values and then picking the top-$K$ items.

**Link Growing**    It then suffices to figure out a way to implement the link growing step of Algorithm 1 (Line 10). Denote by $L$ the scalar loss function of the whole neural network containing layer $l$. Assume that we have just performed a backward propagation, where layer $l$ contributes only $O((1-s)N + n^{(in)} + n^{(out)})$ to the computation time, and $\mathbf{g}^{(x)} := \frac{\partial L}{\partial \mathbf{x}}$, $\mathbf{g}^{(y)} := \frac{\partial L}{\partial \mathbf{y}}$ and $\mathbf{g}^{(\theta)} := \frac{\partial L}{\partial \theta}$ all have been computed. Since $\theta$ is in a compact representation with $(1-s)N$ elements, now $\mathbf{g}^{(\theta)}$ obtained by auto-grad also only contains $(1-s)N$ items, but the growing step Line 10 basically asks to collect the top-$K$ items in the whole *dense* gradient matrix with $N$ elements.

According to the chain rule, for a link $\theta_{ji}$ between the $i$-th input neural and the $j$-th output neural, the partial derivative of $L$ with respect to $\theta_{ji}$ is given by (here, we abuse the notation a little)

$$g_{ji}^{(\theta)} := \frac{\partial L}{\partial \theta_{ji}} = \frac{\partial L}{\partial y_j} \frac{y_j}{\partial \theta_{ji}} = g_j^{(y)} x_i.$$

Hence the desired true dense $n^{(out)} \times n^{(in)}$ gradient matrix $\mathbf{g}^{(\theta)}$ is equal to $\mathbf{g}^{(y)}\mathbf{x}^T$. Our task reduces to collect the $K$ entries with the largest absolute values while keeping away from the locations that have just been dropped in Line 9. In fact, this procedure can be efficiently implemented by scanning via $n^{(out)}$ pointers with the help of a heap (a.k.a., a priority queue), described in the following pseudo-code Algorithm 2.

It can be seen that Algorithm 2 consumes $O(n^{(out)} + n^{(in)} + |U| + (1-s)N)$ heap operations and set operations. If all sets $S$ and $U$ are implemented using binary search trees or hash tables, the costs for each heap operation and each set operation are all within $O(\log N)$. Therefore the total running time of Algorithm 2 is $O((1-s)N \log N)$.

---

**Algorithm 2** Efficient Link Growing

---

**Input:** Input featrue vector $\mathbf{x}$, output gradient vector $\mathbf{g}^{(y)}$, base index set $S_0$, dense matrix size $N$, target sparsity $s$, forbid index set $U$

**Output:** An index set $S$ such that $|S| = (1 - s)N$, $S \supseteq S_0$ and $S \cap U = \emptyset$

1: Initialize $S \leftarrow S_0$.
2: Sort $|x|$ to get a permutation $\sigma_1, \ldots, \sigma_{n^{(in)}}$, such that $|x_{\sigma_1}| \geq |x_{\sigma_2}| \geq \cdots \geq |x_{\sigma_{n^{(in)}}}|$.
3: Create a max-heap $H$ whose elements are 3-tuples, comparing elements according to their first components.
4: **for** $j = 1 \ldots n^{(out)}$ **do**
5:     Append $(|g_j^{(y)} x_{\sigma_1}|, j, 1)$ into $H$.
6: **end for**
7: **while** $|S| < (1 - s)N$ **do**
8:     Pop the top triple $(w, j, i)$ out of $H$.
9:     **if** $(j, \sigma_i) \notin S$ and $(j, \sigma_i) \notin U$ **then**
10:         $S \leftarrow S \cup (j, \sigma_i)$
11:     **end if**
12:     **if** $i < n^{(in)}$ **then**
13:         Append $(|g_j^{(y)} x_{\sigma_{i+1}}|, j, i + 1)$ into $H$.
14:     **end if**
15: **end while**
16: Return $S$.

---

## A.2 Derivation of Eq. (4) in Section 4.2.1

Denote $p_\pi(s_{t+1}, a_{t+1}, \cdots, s_{t+n}, a_{t+n} | s_t, a_t)$ the distribution of the trajectory starting from the current state $s_t$ and action $a_t$ under policy $\pi$. For simplicity, we use $\mathbb{E}_\pi$ to denote $\mathbb{E}_{(s_{t+1}, a_{t+1}, \cdots, s_{t+n}, a_{t+n}) \sim p_\pi(\cdot | s_t, a_t)}$, and use $\mathbb{E}_b$ to denote $\mathbb{E}_{(s_{t+1}, a_{t+1}, \cdots, s_{t+n}, a_{t+n}) \sim p_b(\cdot | s_t, a_t)}$, where $\pi$ and $b$ denote the current policy and the behavior policy, respectively. $Q_\pi(s, a)$ denotes the Q function associated with policy $\pi$ as defined in Eq. (1) in the manuscript, i.e., $Q_\pi(s, a) = \mathbb{E}_\pi \left[ \sum_{i=t}^{T} \gamma^{i-t} r(s_i, a_i) | s_t = s, a_t = a \right]$. We also use $\epsilon(s, a)$ to denote the network fitting error, i.e., $\epsilon(s, a) = Q(s, a; \theta) - Q_\pi(s, a)$.

Subsequently, we have:

$$
\mathbb{E}_b[\mathcal{T}_n(s_t, a_t)] - Q_\pi(s_t, a_t)
$$
$$
= \mathbb{E}_b[\sum_{k=0}^{n-1} \gamma^k r_{t+k} + \gamma^n Q(s_{t+n}, \pi(s_{t+n}); \theta)] - \mathbb{E}_\pi[\sum_{k=0}^{n-1} \gamma^k r_{t+k} + \gamma^n Q_\pi(s_{t+n}, \pi(s_{t+n}))]
$$
$$
= \mathbb{E}_b[\sum_{k=0}^{n-1} \gamma^k r_{t+k} + \gamma^n Q(s_{t+n}, \pi(s_{t+n}); \theta)] - \mathbb{E}_\pi[\sum_{k=0}^{n-1} \gamma^k r_{t+k} + \gamma^n Q(s_{t+n}, \pi(s_{t+n}); \theta)]
$$
$$
+ \mathbb{E}_\pi[\gamma^n (Q(s_{t+n}, \pi(s_{t+n}); \theta) - Q_\pi(s_{t+n}, \pi(s_{t+n})))]
$$
$$
= (\mathbb{E}_b[\mathcal{T}_n(s_t, a_t)] - \mathbb{E}_\pi[\mathcal{T}_n(s_t, a_t)]) + \gamma^n \mathbb{E}_\pi[\epsilon(s_{t+n}, \pi(s_{t+n}))].
$$

The first equality holds in the manuscript by definitions in Eq. (1) and of multi-step TD targets. The second equality holds by firstly adding and then subtracting the term $\mathbb{E}_\pi[\gamma^n(Q(s_{t+n}, a_{t+n}; \theta))]$, and the last equality holds by definitions of $\mathcal{T}_n(s, a)$ and $\epsilon(s, a)$. This decomposition shows that the expected error consists of two parts, i.e., the network fitting error and the policy inconsistency error, which are well handled by our multi-step TD targets with a dynamic-capacity buffer.

## A.3 Algorithm of Dynamic-capacity Buffer in Section 4.2.2

Algorithm 3 presents our formal procedure for dynamically controlling the buffer capacity in Section 4.2.2. For each step, a new transition is inserted into the replay buffer. To avoid the problem of policy inconsistency, we check the buffer per $\Delta_b$ steps and drop the oldest transitions if needed.

Specifically, we first set a hard lower bound $B_{\min}$, and a hard upper bound $B_{\max}$ of the buffer capacity. (i) If the buffer size is below $B_{\min}$, we store all newly collected data samples. (ii) If the amount of buffered transitions has reached $B_{\max}$, the oldest transitions will be replaced by the latest transitions. (iii) When the buffer is in $(B_{\min}, B_{\max})$, for each time, we calculate the policy distance between the oldest behavior policy and the current policy, based on the oldest transitions stored in the buffer. In addition, if the policy distance exceeds the threshold $D_0$, the oldest transitions are discarded. The full algorithm is given in Algorithm 3.

---

**Algorithm 3** Dynamic-Capacity Buffer for off-policy DRL

---

1: $\pi$: current policy
2: $\mathcal{B}$: Replay buffer at training step $t$
3: $\Delta_b$: Buffer checking interval
4: $D_0$: Non-consistency threshold
5: $\rho$: Shrinking ratio
6: $B_{\min}, B_{\max}$: Range of capacity
7: **for** each training step $t$ **do**
8:     Interact with environment and store new transitions in $\mathcal{B}$ *// Add new transitions into the buffer*
9:     Sample mini-batch from $\mathcal{B}$
10:    Training with sampled data *// Depends on the training algorithm*
11:    **if** $t \bmod \Delta_b = 0$ *// Check the buffer periodically* **then**
12:        **if** $|\mathcal{B}| \in (B_{\min}, B_{\max})$ *// The buffer capacity is limited in certain range* **then**
13:            Batch of oldest transitions $(s_i, a_i)$
14:            **if** $\mathcal{D}(\pi) > D_0$ *// Drop the oldest transitions if the policy distance exceeds the threshold* **then**
15:                $\mathcal{B} \leftarrow$ Drop ratio $\rho$ of oldest transitions in $\mathcal{B}$
16:            **end if**
17:        **end if**
18:    **end if**
19: **end for**

---

### A.4 DETAILED ANALYSIS OF DYNAMIC BUFFER

Using a dynamic buffer can reduce the gap between target policy and behavioral policy, as we have shown empirically in Section 5.2. In this section, we give a more detailed analysis of the influence of the dynamic buffer.

We first define the notations used in our analysis.

$p(s'|s, a)$: Environment transition probability

$\rho_{\pi,t}^{(s)}$: State distribution in time $t$ under policy $\pi$

$\rho_{\pi,t}^{(s,a)}$: State-action pair distribution in time $t$ under policy $\pi$

$\mu_\pi(\tau)$: Distribution of trajectory $\tau = (s_t, a_t, s_{t+1}, \cdots, s_{t+n}, a_{t+n})$ under policy $\pi$

$d_\pi$: State-action visitation distribution under policy $\pi$, $d_\pi(s) = \sum_{t=0}^\infty \gamma^t \Pr(s_t = s), a_t \sim \pi(\cdot|s_t)$

Lemma A.1 below first shows that for two trajectory distributions generated by different policies, their KL divergence can be expressed by the KL divergence between these two policies.

**Lemma A.1.** $D_{KL}(\mu_b(\cdot|s_t, a_t)||\mu_\pi(\cdot|s_t, a_t)) = \sum_{k=1}^n \mathbb{E}_{s_{t+k} \sim \rho_{b,t+k}^{(s)}} D_{KL}(b(\cdot|s_{t+k})||\pi(\cdot|s_{t+k}))$

*Proof.* The conditional trajectory distribution can be expressed as:

$$\mu_\pi(\tau|s_t, a_t) = \prod_{k=1}^n \pi(a_{t+k}|s_{t+k})p(s_{t+k}|s_{t+k-1}, a_{t+k-1}).$$

Thus,

$$D_{KL}(\mu_b(\cdot|s_t, a_t)||\mu_\pi(\cdot|s_t, a_t)) = \sum_\tau \mu_b(\tau|s_t, a_t) \log \frac{\mu_b(\tau|s_t, a_t)}{\mu_\pi(\tau|s_t, a_t)}$$

$$= \sum_{\tau} \left[ \mu_b(\tau|s_t, a_t) \log \frac{\prod_{k=1}^{n} b(a_{t+k}|s_{t+k})}{\prod_{k=1}^{n} \pi(a_{t+k}|s_{t+k})} \right]$$

$$= \sum_{k=1}^{n} \sum_{\tau} [\mu_b(\tau) \log \frac{b(a_{t+k}|s_{t+k})}{\pi(a_{t+k}|s_{t+k})}].$$

Note that

$$\mathbb{E}_{\tau \sim \mu_b(\cdot)}[\log \frac{\mu_b(\tau|s_t, a_t)}{\mu_\pi(\tau|s_t, a_t)}] = \mathbb{E}_{(s_{t+k}, a_{t+k}) \sim \rho_{b,t}^{(s,a)}(\cdot)}[\log \frac{\mu_b(\tau|s_t, a_t)}{\mu_\pi(\tau|s_t, a_t)}],$$

then

$$D_{KL}(\mu_b(\cdot|s_t, a_t) || \mu_\pi(\cdot|s_t, a_t)) = \sum_{k=1}^{n} \sum_{(s_{t+k}, a_{t+k})} \rho_{b,t+k}^{(s,a)} \log \frac{b(a_{t+k}|s_{t+k})}{\pi(a_{t+k}|s_{t+k})}$$

$$= \sum_{k=1}^{n} \mathbb{E}_{s_{t+k} \sim \rho_{b,t+k}^{(s)}} D_{KL}(b(\cdot|s_{t+k}) || \pi(\cdot|s_{t+k})).$$

$\square$

Proposition A.2 below shows that the relation between the policy inconsistency error defined in equation 4 and the policy distance. This proposition shows that multi-step TD learning can indeed be more robust with a dynamic buffer.

**Proposition A.2.** *The policy inconsistency error in equation 4 can be upper bounded by*

$$|\mathbb{E}_b[\mathcal{T}_n] - \mathbb{E}_\pi[\mathcal{T}_n]| \leq (\frac{1 - \gamma^n}{1 - \gamma} r_m + \gamma^n Q_m) \sqrt{\frac{1}{2} \sum_{k=1}^{n} \mathbb{E}_{s \sim \rho_{b,t}^{(s)}(\cdot)} D_{KL}(b(\cdot|s) || \pi(\cdot|s))},$$

*where $r_m = \sup r - \inf r, Q_m = \sup Q(s, a; \theta) - \inf Q(s, a; \theta)$.*

*Proof.* Suppose the multi-step TD target is bounded, we have

$$|\mathbb{E}_b[\mathcal{T}_n] - \mathbb{E}_\pi[\mathcal{T}_n]| = |\mathbb{E}_{\tau \sim \mu_b(\cdot|s_t, a_t)}[\mathcal{T}_n] - \mathbb{E}_{\tau \sim \mu_\pi(\cdot|s_t, a_t)}[\mathcal{T}_n]|$$

$$\leq (\sup \mathcal{T}_n - \inf \mathcal{T}_n) D_{TV}(\mu_b(\cdot|s_t, a_t) || \mu_\pi(\cdot|s_t, a_t)).$$

According to Pinsker's inequality,

$$D_{TV}(\mu_b(\cdot|s_t, a_t) || \mu_\pi(\cdot|s_t, a_t)) \leq \sqrt{\frac{D_{KL}(\mu_b(\cdot|s_t, a_t) || \mu_\pi(\cdot|s_t, a_t))}{2}}.$$

Thus,

$$|\mathbb{E}_b[\mathcal{T}_n] - \mathbb{E}_\pi[\mathcal{T}_n]| \leq (\sum_{k=0}^{n-1} \gamma^k r_m + \gamma^n Q_m) \sqrt{\frac{D_{KL}(\mu_b(\cdot|s_t, a_t) || \mu_\pi(\cdot|s_t, a_t))}{2}}.$$

Finally, using Lemma A.1, we obtain

$$|\mathbb{E}_b[\mathcal{T}_n] - \mathbb{E}_\pi[\mathcal{T}_n]| \leq (\frac{1 - \gamma^n}{1 - \gamma} r_m + \gamma^n Q_m) \sqrt{\frac{1}{2} \sum_{k=1}^{n} \mathbb{E}_{s \sim \rho_{b,t}^{(s)}(\cdot)} D_{KL}(b(\cdot|s) || \pi(\cdot|s))}.$$

$\square$

Our next result, Proposition A.3, below shows that the mismatch between $\mathcal{L}(\theta)$ and $\hat{\mathcal{L}}(\theta)$ can be controlled by reducing the KL divergence between the target policy and the behavior policy (complete proof in Appendix A.4. Therefore, one can improve the value estimation by eliminating data inconsistency.

**Proposition A.3.** *For the target policy $\pi$ and behavioural policy $b$, we have*

$$|\mathcal{L}(\theta) - \hat{\mathcal{L}}(\theta)| \leq \frac{2\gamma}{1-\gamma}\Delta\mathbb{E}_{s\sim d_b}\left[\sqrt{\frac{1}{2}D_{KL}(\pi(\cdot|s), b(\cdot|s))}\right],$$

*where $\Delta = \sup|(Q(s,a;\theta) - \mathcal{T}(s,a))^2|$ and the loss functions are defined as*

$$\mathcal{L}(\theta) = \mathbb{E}_{(s_i,a_i)\sim d_\pi}[(Q(s_i,a_i;\theta) - \mathcal{T}(s_i,a_i))^2], \hat{\mathcal{L}}(\theta) = \mathbb{E}_{(s_i,a_i)\sim d_b}[(Q(s_i,a_i;\theta) - \mathcal{T}(s_i,a_i))^2].$$

*Proof.* Denote

$$\Delta = \sup|(Q(s,a;\theta) - \mathcal{T}(s,a))^2|.$$

Then, we have

$$|\mathcal{L}(\theta) - \hat{\mathcal{L}}(\theta)| = |\mathbb{E}_{(s_i,a_i)\sim d_b}[(Q(s_i,a_i;\theta) - \mathcal{T}(s_i,a_i))^2] - \mathbb{E}_{(s_i,a_i)\sim d_\pi}[(Q(s_i,a_i;\theta) - \mathcal{T}(s_i,a_i))^2]|$$
$$\leq 2D_{\mathrm{TV}}(d_\pi, d_b)\Delta,$$

i.e., the gap between the two loss functions can be bounded by the total variance distance between the two state-action visitation distributions.

According to Achiam et al. (2017), we obtain

$$D_{\mathrm{TV}}(d_\pi, d_b) \leq \frac{\gamma}{1-\gamma}\mathbb{E}_{s\sim d_b}[D_{\mathrm{TV}}(\pi(\cdot|s), b(\cdot|s))].$$

Thus, the loss function gap can be bounded by the total variance distance between the two policies, i.e.,

$$|\mathcal{L}(\theta) - \hat{\mathcal{L}}(\theta)| \leq \frac{2\gamma}{1-\gamma}\Delta\mathbb{E}_{s\sim d_b}[D_{\mathrm{TV}}(\pi(\cdot|s), b(\cdot|s))].$$

With Pinsker's inequality, we can express the upper bound with the KL divergence between the two policies:

$$|\mathcal{L}(\theta) - \hat{\mathcal{L}}(\theta)| \leq \frac{2\gamma}{1-\gamma}\Delta\mathbb{E}_{s\sim d_b}[\sqrt{\frac{1}{2}D_{\mathrm{KL}}(\pi(\cdot|s), b(\cdot|s))}].$$

$\square$

# B  DETAILS OF RLX2 WITH TD3 AND SAC

In this section, we provide the pseudo-codes of instantiations of RLx2 on TD3 and SAC in Algorithm 4 and Algorithm 5, respectively. We emphasize that RLx2 is a general sparse training framework for off-policy DRL and can be applied to training other DRL algorithms apart from TD3 and SAC, with sparse networks from scratch. Below, we first illustrate the critical steps of RLx2 in Algorithm 4, using TD3 as the base algorithm.

**Topology evolution** is performed in Lines 15-17 and Lines 20-22 in Algorithm 1. Specifically, we first calculate the sparsity of each layer according to the target sparsity of the total model at initialization. The sparsity of each layer is fixed during the training. And we use the Erdős–Rényi strategy, which is introduced in Mocanu et al. (2018), to allocate the sparsity to each layer. For a sparse network with $L$ layers, this strategy utilizes the equations below:

$$(1-S)\sum_l I_l O_l = \sum_{l=1}^{L}(1-S_l)I_l O_l,$$
$$1-S_l = k\frac{I_l + O_l}{I_l * O_l},$$

where $S$ is the target sparsity of the model, $S_l$ is the sparsity of the $l$-th layer, $I_l$ is the input dimensionality of the $l$-th layer, $O_l$ is the output dimensionality of the $l$-th layer, and $k$ is a constant. The motivation of this strategy is that a layer with more parameters contains more redundancy. As

a result, it can be compressed with a higher sparsity. The topology evolution update is performed every $\Delta_m$ time steps. Definitions of other hyperparameters related to topology evolution are listed in Algorithm 1 in the manuscript.

**Buffer capacity adjustment** is performed in Lines 7-9. This adjustment is conducted every $\Delta_b$ step, with detailed procedure shown in Algorithm 3.

**Multi-step TD target** is computed in Lines 10-13. We found that using a multi-step TD target in the early stage of training may result in poor performance because the policy may evolve quickly, which results in severe policy inconsistency. Thus, we start the multi-step TD target only when the number of training steps succeeds a pre-set threshold $T_0$. As mentioned in Section 4.2.1, the one-step TD target and multi-step TD target in TD3 are computed as:

$$\mathcal{T}_1 = r_t + \gamma Q(s_{t+1}, \pi(s_{t+1}); \theta),$$
$$\mathcal{T}_n = \sum_{k=0}^{n-1} \gamma^k r_{t+k} + \gamma^n Q(s_{t+n}, \pi(s_{t+n}); \theta). \tag{7}$$

Note that the calculation of the multi-step TD target in SAC is slightly different from that in TD3. Specifically, the one-step TD target for SAC is computed as:

$$\mathcal{T}_1 = r_t + \gamma(Q(s_{t+1}, \tilde{a}_{t+1}; \theta) - \alpha \log \pi(\tilde{a}_{t+1}|s_{t+1})), \tag{8}$$

where $\tilde{a}_{t+1} \sim \pi(\cdot|s_{t+1})$, and the $n$-step TD target for SAC is computed by

$$\mathcal{T}_n = \sum_{k=0}^{n-1} \gamma^k r_t + \gamma^n Q(s_{t+n}, \tilde{a}_{t+n}; \theta) - \alpha \sum_{k=0}^{n-1} \gamma^{k+1} \log \pi(\tilde{a}_{t+k+1}|s_{t+k+1}), \tag{9}$$

where $\tilde{a}_{t+k+1} \sim \pi(\cdot|s_{t+k+1}), k = 0, 1, \cdots, n-1$. Due to this difference, we will see later in Section C.3 that the resulting FLOPs are slightly different for TD3 and SAC.

Besides, Algorithm 5 also gives the detailed implementation of RLx2 with SAC, where topology evolution is performed in Lines 15-17 and Lines 20-22, buffer capacity adjustment is performed in Lines 7-9, and the multi-step TD target is computed in Lines 10-13.

---

**Algorithm 4** RLx2-TD3

---

1: Initialize sparse critic network $Q_{\theta_1}$, $Q_{\theta_2}$ and sparse actor network $\pi_\phi$ with random parameters $\theta_1$, $\theta_2$, $\phi$ and random masks $M_{\theta_1}$, $M_{\theta_2}$, $M_\phi$ with determined sparsity $S^{(c)}$, $S^{(a)}$.
2: $\theta_1 \leftarrow \theta_1 \odot M_{\theta_1}, \theta_2 \leftarrow \theta_2 \odot M_{\theta_2}, \phi \leftarrow \phi \odot M_\phi$. *// Start with a random sparse network*
3: Initialize target networks $\theta_1' \leftarrow \theta_1, \theta_2' \leftarrow \theta_2, \phi' \leftarrow \phi$. Initialize replay buffer $\mathcal{B}$.
4: **for** $t = 1$ to $T$ **do**
5:     Select action with exploration noise $a_t \sim \pi_\phi(s_t) + \epsilon, \epsilon \sim \mathcal{N}(0, \sigma)$ and observe reward $r_t$ and new state $s_{t+1}$
6:     Store transition tuple $(s_t, a_t, r_t, s_{t+1})$ in $\mathcal{B}$
7:     **if** $t \bmod \Delta_b = 0$ **then**
8:         Buffer capacity adjustment
9:     **end if** *// Check the buffer periodically*
10:     Set $N = 1$ temporarily if $t < T_0$ *// Delay to use multi-step TD target*
11:     Sample mini-batch of $B$ multi-step transitions $(s_i, a_i, r_i, s_{i+1}, a_{i+1}, \cdots, s_{i+N})$ from $\mathcal{B}$
12:     $\tilde{a} \leftarrow \pi_{\phi'}(s_{i+N}) + \epsilon, \epsilon \sim clip(\mathcal{N}(0, \tilde{\sigma}), -c, c)$
13:     Calculate multi-step TD target $y \leftarrow \sum_{k=0}^{N-1} \gamma^k r_{i+k} + \gamma^N \min_{j=1,2} Q_{\theta_j'}(s_{i+N}, \tilde{a})$ *// multi-step TD target*
14:     Update critic networks $\theta_j \leftarrow \theta_j - \lambda \nabla_{\theta_j} \frac{1}{B} \sum (y - Q_{\theta_j}(s_i, a_i))^2$ for $j = 1, 2$
15:     **if** $t \bmod \Delta_m = 0$ **then**
16:         Topology_Evolution($Q_{\theta_j}$) for $j = 1, 2$
17:     **end if** *// Update the mask of critic periodically*
18:     **if** $t \bmod d = 0$ **then**
19:         Update actor network $\phi \leftarrow \phi - \lambda \nabla_\phi (-\frac{1}{B} \sum Q_{\theta_1}(s_i, a_i))$
20:         **if** $t/d \bmod \Delta_m = 0$ **then**
21:             Topology_Evolution($\pi_\phi$)
22:         **end if** *// Update the mask of actor periodically*
23:         Update target networks:
        $\theta_i' \leftarrow \tau\theta_i + (1-\tau)\theta_i', \phi' \leftarrow \tau\phi + (1-\tau)\phi',$
        $\theta_i' \leftarrow \theta_i' \odot M_{\theta_i}, \phi' \leftarrow \phi' \odot M_\phi$ *// Target networks are also sparsified with the same mask*
24:     **end if**
25: **end for**

---

---

**Algorithm 5** RLx2-SAC

---

1: Initialize sparse critic network $Q_{\theta_1}$, $Q_{\theta_2}$ and sparse actor network $\pi_\phi$ with random parameters $\theta_1$, $\theta_2$, $\phi$ and random masks $M_{\theta_1}$, $M_{\theta_2}$, $M_\phi$ with determined sparsity $S^{(c)}$, $S^{(a)}$.
2: $\theta_1 \leftarrow \theta_1 \odot M_{\theta_1}$, $\theta_2 \leftarrow \theta_2 \odot M_{\theta_2}$, $\phi \leftarrow \phi \odot M_\phi$. *// Start with a random sparse network*
3: Initialize target networks $\theta'_1 \leftarrow \theta_1$, $\theta'_2 \leftarrow \theta_2$. Initialize replay buffer $\mathcal{B}$.
4: **for** $t = 1$ to $T$ **do**
5:     Select action $a_t \sim \pi_\phi(s_t)$, and observe reward $r_t$ and new state $s_{t+1}$
6:     Store transition tuple $(s_t, a_t, r_t, s_{t+1})$ in $\mathcal{B}$
7:     **if** $t \mod \Delta_b = 0$ **then**
8:         Buffer capacity adjustment
9:     **end if** *// Check the buffer periodically*
10:    Set $N = 1$ temporarily if $t < T_0$ *// Delay to use multi-step TD target*
11:    Sample mini-batch of $B$ multi-step transitions $(s_i, a_i, r_i, s_{i+1}, a_{i+1}, \cdots, s_{i+N})$ from $\mathcal{B}$
12:    $\tilde{a}_{i+k+1} \sim \pi(\cdot|s_{i+k+1}), k = 0, 1, \cdots, N - 1$
13:    Calculate multi-step TD target *// multi-step TD target*
        $y \leftarrow \sum_{k=0}^{N-1} \gamma^k r_{i+k} + \gamma^N \min_{j=1,2} Q_{\theta'_j}(s_{i+N}, \tilde{a}_{i+N}) - \alpha \sum_{k=0}^{N-1} \gamma^{k+1} \log \pi(\tilde{a}_{i+k+1}|s_{i+k+1})$
14:    Update critic networks $\theta_j \leftarrow \theta_j - \lambda \nabla_{\theta_j} \frac{1}{B} \sum (y - Q_{\theta_j}(s_i, a_i))^2$ for $j = 1, 2$
15:    **if** $t \mod \Delta_m = 0$ **then**
16:        Topology_Evolution($Q_{\theta_j}$) for $j = 1, 2$
17:    **end if** *// Update the mask of critic periodically*
18:    Update actor network $\phi \leftarrow \phi - \lambda \nabla_\phi (-\frac{1}{B} \sum \min_{j=1,2} Q_{\theta_j}(s_i, a_i))$
19:    **if** $t/d \mod \Delta_m = 0$ **then**
20:        Topology_Evolution($\pi_\phi$)
21:    **end if** *// Update the mask of actor periodically*
22:    Automating Entropy Adjustment: $\alpha \leftarrow \alpha - \lambda \nabla_\alpha \frac{1}{B} \sum (-\alpha \log \pi(a_i|s_i) - \alpha \overline{\mathcal{H}})$
23:    Update target networks:
        $\theta'_i \leftarrow \tau \theta_i + (1 - \tau)\theta'_i, \theta'_i \leftarrow \theta'_i \odot M_{\theta_i}$ *// Target networks are also sparsified with the same mask*
24: **end for**

---

## C    EXPERIMENTAL DETAILS

We provide more experimental details in this section, including the detailed experimental setup, the calculations of model size and FLOPs, and supplementary experiment results.

### C.1    HARDWARE SETUP

Our experiments are implemented with PyTorch (Paszke et al., 2017) and run on 8x P100 GPUs. Each run needs 12 hours for TD3 and 2 days for SAC for three million steps. The code will be open-sourced upon publication of the paper.

### C.2    HYPERPARAMETER SETTINGS FOR REPRODUCTION

Table 4 presents detailed hyperparameters of RLx2-TD3 and RLx2-SAC in our experiments.

### C.3    CALCULATION OF MODEL SIZE AND FLOPS

We present the details of calculating model sizes and FLOPs in this subsection, where focus on fully-connected layers since the networks used in our experiments are all Multilayer Perceptrons (MLPs). These calculations can be easily extended to convolutional layers or other architectures. Besides, we omit the offset term in fully-connected layers in our calculations.

### C.3.1    MODEL SIZE

We first illustrate the calculation of model sizes, i.e., the total number of parameters in the model. Initially, for a sparse network with $L$ fully-connected layers, we calculate the model size as:

$$M = \sum_{l=1}^{L} (1 - S_l) I_l O_l,$$

Table 4: Hyperparameters of RLx2-TD3 and RLx2-SAC.

| Category | Hyperparameter | Value |
|---|---|---|
| Shared Hyperparameters | Optimizer | Adam |
| | Learning rate $\lambda$ | $3 \times 10^{-4}$ |
| | Discount factor $\gamma$ | 0.99 |
| | Number of hidden layers (all networks) | 2 |
| | Number of hidden units per layer | 256 |
| | Activation Function | ReLU |
| | Batch size $B$ | 256 |
| | Warmup steps | 25000 |
| | Target update rate $\tau$ | 0.005 |
| | Initial mask update fraction $\zeta_0$ | 0.5 |
| | Mask update interval $\Delta_m$ | 10000 |
| | Buffer adjustment interval $\Delta_b$ | 10000 |
| | Buffer policy distance threshold $D_0$ | 0.2 |
| | Buffer max size $B_{\max}$ | $1 \times 10^6$ |
| | Buffer min size $B_{\min}$ | $1 \times 10^5$ |
| | Multi-step delay $T_0$ | $3 \times 10^5$ |
| Hyperparameters for RLx2-TD3 | Target update interval | 2 |
| | Actor update interval $d$ | 2 |
| | Exploration policy | $\mathcal{N}(0, 0, 1)$ |
| | Multi-step | 3 |
| Hyperparameters for RLx2-SAC | Target update interval | 1 |
| | Actor update interval $d$ | 1 |
| | Entropy target $\overline{\mathcal{H}}$ | $-\dim(\mathcal{A})$ |
| | Multi-step | 2 |

where $S_l$ is the sparsity, $I_l$ is the input dimensionality, and $O_l$ is the output dimensionality of the $l$-th layer. Specifically, the "Total Size" column in Table 2 in the manuscript refers the model size including both actor and critic networks during training.

For both TD3 and SAC, double Q-learning is adopted, i.e., train two value networks concurrently. Besides, we also use target networks in our implementations as target critics for both TD3 and SAC, yet a target actor only for TD3. Thus, if we denote $M_{\text{Actor}}$ and $M_{\text{Critic}}$ as model sizes of actor and critic, respectively, the detailed calculation of model sizes can be obtained as shown in the second column of Table 5. We denote $B$ as the batch size used for training process.

Table 5: FLOPs and model size for RLx2-TD3 and RLx2-SAC.

| Algorithm | Model size | FLOPs (average of each iteration during training) | FLOPs (inference) |
|---|---|---|---|
| RLx2-TD3 | $2M_{\text{Actor}} + 4M_{\text{Critic}}$ | $B \times (2.5\text{FLOPs}_{\text{Actor}} + 8.5\text{FLOPs}_{\text{Critic}})$ | $\text{FLOPs}_{\text{Actor}}$ |
| RLx2-SAC | $M_{\text{Actor}} + 4M_{\text{Critic}}$ | $B \times (5\text{FLOPs}_{\text{Actor}} + 10\text{FLOPs}_{\text{Critic}})$ | $\text{FLOPs}_{\text{Actor}}$ |

### C.3.2 FLOPs

Initially, for a sparse network with $L$ fully-connected layer, the required FLOPs for a forward pass is competed as follows (also adopted in Evci et al. (2020) and Molchanov et al. (2019a)):

$$\text{FLOPs} = \sum_{l=1}^{L} (1 - S_l)(2I_l - 1)O_l, \tag{10}$$

where $S_l$ is the sparsity, $I_l$ is the input dimensionality, and $O_l$ is the output dimensionality of the $l$-th layer. Denote $\text{FLOPs}_{\text{Actor}}$ and $\text{FLOPs}_{\text{Critic}}$ as the FLOPs required in a forward pass of a single

actor and critic network, respectively. The inference FLOPs is exactly FLOPs$_\text{Actor}$ as shown in the last column of Table 5. As for the training FLOPs, the calculation consisted of multiple forward and backward passes in several networks, which will be detailed below.

In particular, we compute the FLOPs needed for each training iteration. Besides, we omit the FLOPs of the following processes since they have little influence on the final result.

(i) *Interaction with the environment.* Each time the agent decides an action to interact with the environment takes FLOPs as FLOPs$_\text{Actor}$, which is much smaller than the FLOPs need for updating networks as shown in Table 5 since $B \gg 1$.

(ii) *Updating target networks.* Every parameter in the networks is updated as $\theta' \leftarrow \tau\theta + (1-\tau)\theta'$. Thus, the number of FLOPs here is the same as the model size, which is also negligible.

(iii) *Topology evolution and buffer capacity adjustment.* These two components are performed every 10000 steps. Formally speaking, the average FLOPs of topology evolution is give by $B \times \frac{2\text{FLOPs}_\text{Actor}}{(1-S^{(a)})\Delta_m}$ for the actor, and $B \times \frac{4\text{FLOPs}_\text{Critic}}{(1-S^{(c)})\Delta_m}$ for the critic, where $S^{(a)}$ and $S^{(c)}$ are the sparsity of actor and critic, respectively. The FLOPs of buffer capacity adjustment is $8B \times \frac{\text{FLOPs}_\text{Actor}}{\Delta_b}$, where $8B$ is because that we use the oldest $8B$ transitions to compute the policy distance. Thus, they are both negligible.

Therefore, we focus on the FLOPs of updating actor and critic. The average FLOPs of updating actor and critic can be given as:

$$\text{FLOPs}_\text{train} = \text{FLOPs}_\text{update\_critic} + \frac{1}{d}\text{FLOPs}_\text{update\_actor}, \tag{11}$$

where $d$ is the actor update interval (2 for TD3 and 1 for SAC in our implementations). Next we calculate the FLOPs of updating actor and critic, i.e. FLOPs$_\text{update\_critic}$ and FLOPs$_\text{update\_actor}$. We first focus on TD3, while that for SAC is similar.

**Training FLOPs Calculation in TD3**
(i) *Critic FLOPs:* Recall the way to update critic (two critics, $\theta_1$ and $\theta_2$) in TD3 is given by

$$\theta_j \leftarrow \theta_j - \lambda\nabla_{\theta_j}\frac{1}{B}\sum(\mathcal{T}_n - Q(s_i, a_i; \theta_j))^2, \tag{12}$$

for $j = 1, 2$, where $B$ is the batch size, $n$-step TD target

$$\mathcal{T}_n = \sum_{k=0}^{N-1}\gamma^k r_{i+k} + \gamma^N \min_{j=1,2} Q(s_{i+N}, \tilde{a}; \theta'_j) \tag{13}$$

and $\theta'_j$ refers to the target network.

Subsequently, we can compute the FLOPs of updating critic as:

$$\text{FLOPs}_\text{update\_critic} = \text{FLOPs}_\text{TD\_target} + \text{FLOPs}_\text{compute\_loss} + \text{FLOPs}_\text{backward\_pass}, \tag{14}$$

where FLOPs$_\text{TD\_target}$, FLOPs$_\text{compute\_loss}$, and FLOPs$_\text{backward\_pass}$ refer to the numbers of FLOPs in computing the TD targets in forward pass, loss function in forward pass, and gradients in backward pass (backward-propagation), respectively. By Eq. (12) and Eq. (13) we have:

$$\text{FLOPs}_\text{TD\_target} = B \times (\text{FLOPs}_\text{Actor} + 2\text{FLOPs}_\text{Critic}),$$
$$\text{FLOPs}_\text{compute\_loss} = B \times 2\text{FLOPs}_\text{Critic}. \tag{15}$$

For the FLOPs of gradients backward propagation, FLOPs$_\text{backward\_pass}$, we compute it as two times the computational expense of the forward pass, which is adopted in existing literature (Evci et al., 2020), i.e.,

$$\text{FLOPs}_\text{backward\_pass} = B \times \underline{2} \times 2\text{FLOPs}_\text{Critic}, \tag{16}$$

where the extra factor $\underline{2}$ comes from the cost of double Q-learning.

Combining Eq. (14), Eq. (15), and Eq. (16), the FLOPs of updating critic in TD3 is:

$$\text{FLOPs}_\text{update\_critic} = B \times (\text{FLOPs}_\text{Actor} + 8\text{FLOPs}_\text{Critic}). \tag{17}$$

(ii) *Actor FLOPs:* Recall the way to update actor (parameterized by $\phi$) in TD3 is given by

$$\phi \leftarrow \phi - \lambda\nabla_\phi(-\frac{1}{B}\sum Q(s_i, a_i; \theta_1)),$$

where $\theta_1$ refers to a critic network. Subsequently, we compute the FLOPs of updating actor as:

$$\text{FLOPs}_{\text{update\_actor}} = \text{FLOPs}_{\text{compute\_loss}} + \text{FLOPs}_{\text{backward\_pass}}, \tag{18}$$

and similar to the calculations of updating critic, we have:

$$\begin{aligned}
\text{FLOPs}_{\text{compute\_loss}} &= B \times (\text{FLOPs}_{\text{Actor}} + \text{FLOPs}_{\text{Critic}}), \\
\text{FLOPs}_{\text{backward\_pass}} &= B \times 2\text{FLOPs}_{\text{Actor}}.
\end{aligned} \tag{19}$$

Combining Eq. (18), Eq. (18) and Eq. (19), the flops of updating actor in TD3 is:

$$\text{FLOPs}_{\text{update\_actor}} = B \times (3\text{FLOPs}_{\text{Actor}} + \text{FLOPs}_{\text{Critic}}) \tag{20}$$

**Training FLOPs Calculation in SAC**
(i) *Critic FLOPs:* Calculations of FLOPs for SAC are similar to that in TD3. The way to update the critic in SAC is:

$$\theta_j \leftarrow \theta_j - \lambda \nabla_{\theta_j} \frac{1}{B} \sum (\mathcal{T}_n - Q_{\theta_j}(s_i, a_i))^2$$

for $j = 1, 2$, where $B$ is the batch size, $n$-step TD target

$$\mathcal{T}_n = \sum_{k=0}^{N-1} \gamma^k r_{i+k} + \gamma^N \min_{j=1,2} Q_{\theta'_j}(s_{i+N}, \tilde{a}_{i+N}) - \alpha \sum_{k=0}^{N-1} \gamma^{k+1} \log \pi(\tilde{a}_{i+k+1}|s_{i+k+1})$$

and $\theta'_j$ refers to the target network. Note that the way to compute multi-step TD target in SAC is slightly different from which in TD3, we have:

$$\text{FLOPs}_{\text{TD\_target}} = B \times (2\text{FLOPs}_{\text{Actor}} + 2\text{FLOPs}_{\text{Critic}}). \tag{21}$$

Other terms for updating critic are the same as those in Eq. (14) for TD3. Thus, the FLOPs of updating critic in SAC can be computed by

$$\text{FLOPs}_{\text{update\_critic}} = B \times (2\text{FLOPs}_{\text{Actor}} + 8\text{FLOPs}_{\text{Critic}}). \tag{22}$$

(ii) *Actor FLOPs:* The way to update the actor in SAC is:

$$\phi \leftarrow \phi - \lambda \nabla_\phi (-\frac{1}{B} \sum \min_{j=1,2} Q_{\theta_j}(s_i, a_i)),$$

where $\theta_j$ refers to a critic network. Subsequently, we have:

$$\text{FLOPs}_{\text{compute\_loss}} = B \times (\text{FLOPs}_{\text{Actor}} + 2\text{FLOPs}_{\text{Critic}}). \tag{23}$$

The backward pass FLOPs is the same as that in TD3, i.e.,

$$\text{FLOPs}_{\text{backward\_pass}} = B \times 2\text{FLOPs}_{\text{Actor}}. \tag{24}$$

Thus, the FLOPs of updating the actor in SAC is:

$$\text{FLOPs}_{\text{update\_actor}} = B \times (3\text{FLOPs}_{\text{Actor}} + 2\text{FLOPs}_{\text{Critic}}). \tag{25}$$

Table 6 shows the relative average FLOPs of each iteration for different algorithms, where the FLOPs of training a sparse network without any of these three methods is set to 1x. The sparsity is set to the average sparsity in different environments. The additional computations induced by topology evolution and dynamic buffer are negligible. Using multi-step TD learning also does not increase computations in TD3, and only introduces a small extra computation to SAC ($< 5\%$), as analyzed above.

Table 6: Relative FLOPs of different methods

| Algorithm | Tiny | SS | SET | RigL | RLx2 |
|-----------|------|-----|------|------|------|
| TD3 | 1.00000x | 1.00000x | 1.00000x | 1.00067x | 1.00072x |
| SAC | 1.00000x | 1.00000x | 1.00000x | 1.00033x | 1.04432x |

## C.4    COMPARISONS BETWEEN DIFFERENT MASKS IN OTHER ENVIRONMENTS

Figure 8 shows the improvement of the learned sparse network topology by robust value learning. We see that in addition to Ant-v3, significant improvements are also achieved in Hopper-v3 and Walker2d-v3. The only exception is HalfCheetah-v3. We hypothesize that this environment is easier for the agent to learn (as the performance improves much faster in the early stage than in the other three environments), and a good mask can be found comparatively easier. It is an interesting direction for future work to systematically analyze this problem.

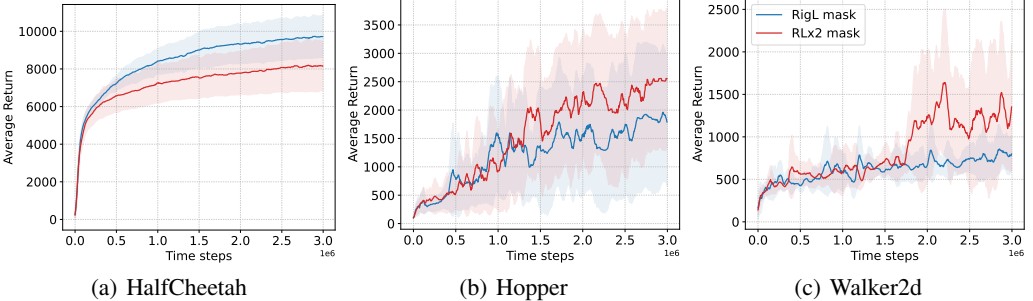

(a) HalfCheetah    (b) Hopper    (c) Walker2d

Figure 8: Comparison of different sparse topologies learned in HalfCheetah-v3, Hopper-v3, and Walker2d-v3.

## C.5    TRAINING CURVES OF COMPARATIVE EVALUATION IN SECTION 5.1

Figure 9 and Figure 10 show the training curves of different algorithms in four MuJoCo environments. RLx2 outperforms baseline algorithms on all four environments with both TD3 and SAC.

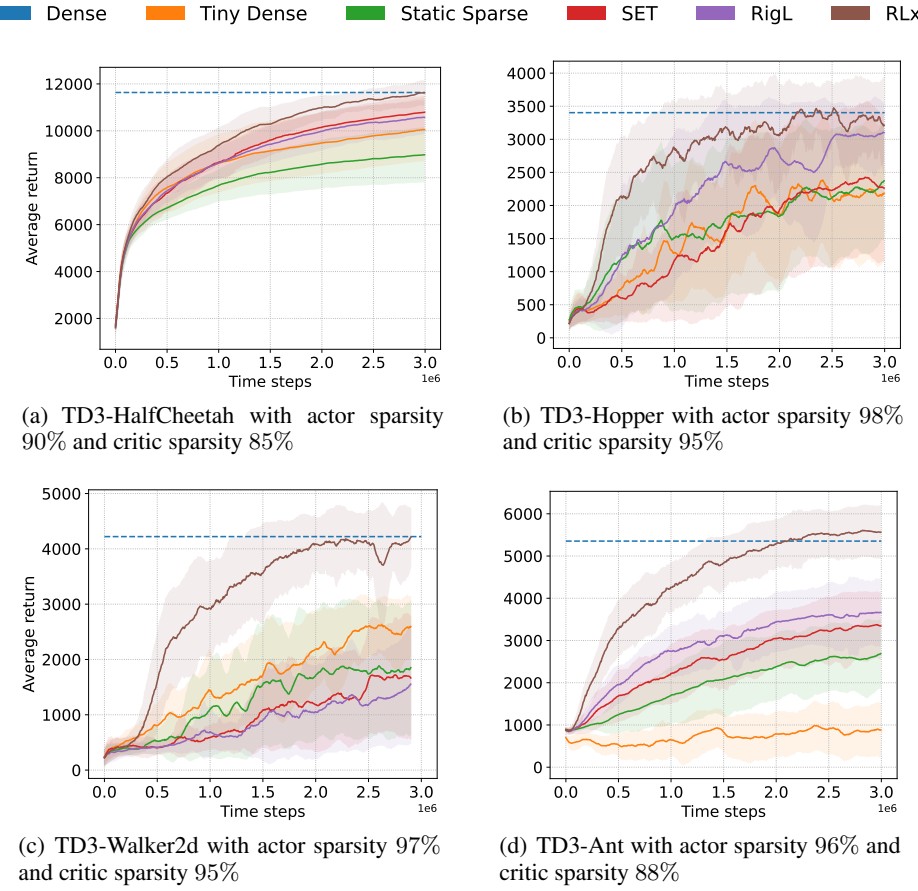

(a) TD3-HalfCheetah with actor sparsity 90% and critic sparsity 85%

(b) TD3-Hopper with actor sparsity 98% and critic sparsity 95%

(c) TD3-Walker2d with actor sparsity 97% and critic sparsity 95%

(d) TD3-Ant with actor sparsity 96% and critic sparsity 88%

Figure 9: Training processes of RLx2-TD3 on four MuJoCo environments. The performance is calculated as the average reward per episode over the last 30 evaluations of the training.

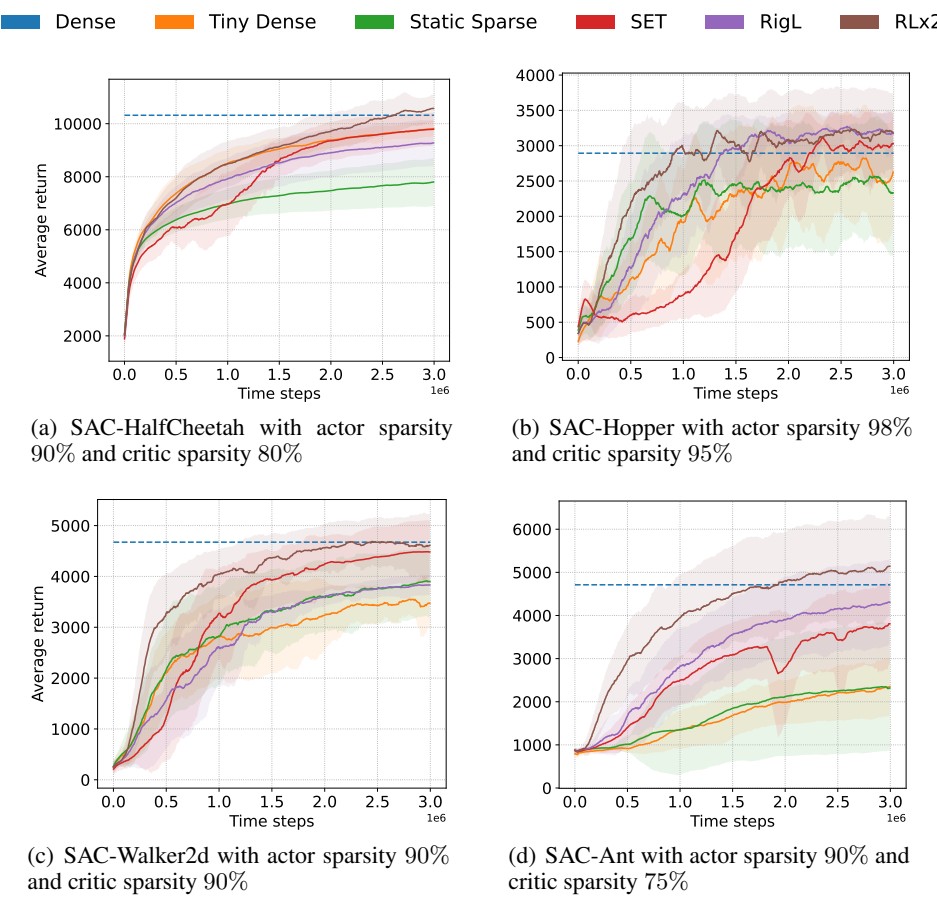

(a) SAC-HalfCheetah with actor sparsity 90% and critic sparsity 80%

(b) SAC-Hopper with actor sparsity 98% and critic sparsity 95%

(c) SAC-Walker2d with actor sparsity 90% and critic sparsity 90%

(d) SAC-Ant with actor sparsity 90% and critic sparsity 75%

Figure 10: Training processes of RLx2-SAC on four MuJoCo environments. The performance is calculated as the average reward per episode over the last 30 evaluations of the training.

## C.6 STANDARD DEVIATIONS OF RESULTS IN TABLE 2

Table 7 shows the performance of different algorithms on four MuJoCo environments with standard deviations. Each result is calculated on 8 random seeds. RLx2 does not lead to a larger variance with topology evolution.

Table 7: Results in Table 2 with standard deviations

| Alg. | Env. | Tiny(%) | SS(%) | SET(%) | RigL(%) | RLx2(%) |
|---|---|---|---|---|---|---|
| TD3 | Hal. | 86.3±11.6 | 77.1±10.1 | 92.6±6.1 | 90.8±6.3 | **99.8**±4.7 |
| | Hop. | 64.5±31.1 | 67.7±26.0 | 66.5±33.0 | 90.6±14.9 | **97.0**±17.6 |
| | Wal. | 60.8±13.1 | 42.9±27.8 | 39.3±26.1 | 35.7±24.3 | **98.1**±15.2 |
| | Ant. | 16.5±11.9 | 49.6±15.8 | 62.5±14.8 | 68.5±15.0 | **103.9**±11.9 |
| | Avg. | 57.0±16.9 | 59.3±20.0 | 65.2±20.0 | 71.4±15.1 | **99.7**±12.4 |
| SAC | Hal. | 95.0±2.7 | 75.4±8.7 | 94.8±6.0 | 89.8±8.4 | **102.2**±3.2 |
| | Hop. | 89.1±28.0 | 81.6±30.6 | 103.9±15.7 | **110.0**±10.4 | 109.7±19.9 |
| | Wal. | 73.8±11.4 | 83.4±14.8 | 95.8±13.1 | 81.9±4.3 | **103.2**±13.3 |
| | Ant | 49.6±14.3 | 49.3±31.0 | 79.8±20.8 | 90.9±21.3 | **105.6**±24.7 |
| | Avg. | 76.9±14.1 | 72.4±21.2 | 93.6±13.9 | 93.2±11.1 | **105.2**±15.3 |
| | Avg. | 67.0±15.5 | 65.9±20.6 | 79.4±17.0 | 82.3±13.1 | **101.8**±13.9 |

## C.7 Supplementary Results for Ablation Study

Table 8 shows the performance of four environments with different buffer capacities. Consistent with the results in Section 5.2, a buffer that is either too small or too large can result in poor performance. Our dynamic buffer outperforms buffers with a fixed capacity.

Table 8: RLx2-TD3 with different buffer capacity.

| Environment | Capacity $5 \times 10^4$ | Capacity $1 \times 10^5$ | Capacity $2 \times 10^5$ | Capacity $1 \times 10^6$ | Unlimited | Dynamic |
|---|---|---|---|---|---|---|
| HalfCheetah | 84.5% | 86.2% | 93.7% | 95.6% | 88.2% | **99.8%** |
| Hopper | 79.9% | 94.1% | 91.4% | 94.5% | 89.1% | **97.0%** |
| Walker2d | 77.9% | 94.7% | 90.9% | 90.2% | 79.2% | **98.1%** |
| Ant | 75.6% | 75.1% | 83.4% | 80.0% | 72.6% | **103.9%** |
| Average | 79.5% | 87.5% | 89.9% | 90.1% | 82.3% | 99.7% |

## C.8 Sensitivity Analysis for Hyperparameters

In this section, we provide the detailed sensitivity analysis for new hyperparameters used in RLx2, including initial mask update fraction $\zeta$, mask update interval $\Delta_m$, buffer adjustment interval $\Delta_b$, buffer policy distance threshold $D_0$, and multi-step delay $T_0$.

**Initial mask update fraction** Table 9 shows the performance with different initial mask update fractions (denoted as $\zeta$) in different environments. We also include the special case of keeping the mask static, i.e., $\zeta = 0$. From Table 9, we find that the sensitivities of the initial mask update fraction among different environments are similar. Besides, RLx2 achieves better performance with a large value of the initial mask update fraction. There is no apparent performance degradation even if $\zeta$ is set to 0.9, which may be due to the update fraction annealing scheme.

Table 9: Sensitivity analysis on initial mask update fraction.

| Environment | Static Sparse | $\zeta = 0.1$ | $\zeta = 0.3$ | $\zeta = 0.5$ | $\zeta = 0.7$ | $\zeta = 0.9$ |
|---|---|---|---|---|---|---|
| HalfCheetah | 86.1% | 93.5% | 96.2% | 99.8% | 100.5% | 103.5% |
| Hopper | 84.2% | 98.2% | 103.7% | 97.0% | 97.0% | 92.4% |
| Walker2d | 83.8% | 98.2% | 96.4% | 98.1% | 104.3% | 101.4% |
| Ant | 80.2% | 79.1% | 83.8% | 103.9% | 101.9% | 102.8% |
| Average | 83.6% | 92.3% | 95.0% | 99.7% | 100.9% | 100.0% |

**Mask update interval** Table 10 shows the performance with different mask update intervals (denoted as $\Delta_m$) in different environments. We also include the special case of never updating the mask, i.e., $\Delta_m = \infty$. From Table 10, we find that the sensitivities of the mask update interval among different environments are similar. Table 10 also shows that a small mask update interval reduces the performance since adjusting the mask too frequently may drop the critical connections before their weights are updated to large values by the optimizer. On the contrary, a large mask update interval reduces the impact caused by topology evolution but degrades to training a static sparse network. In general, a moderate value of $1 \times 10^4$ is favoured.

Table 10: Sensitivity analysis on mask update interval.

| Environment | $\Delta_m = 1 \times 10^3$ | $\Delta_m = 3 \times 10^3$ | $\Delta_m = 1 \times 10^4$ | $\Delta_m = 3 \times 10^4$ | $\Delta_m = 1 \times 10^5$ | Static $\Delta_m = \infty$ |
|---|---|---|---|---|---|---|
| HalfCheetah | 96.4% | 93.7% | 99.8% | 100.8% | 94.6% | 86.1% |
| Hopper | 97.1% | 98.7% | 97.0% | 91.8% | 97.8% | 84.2% |
| Walker2d | 98.2% | 103.7% | 98.1% | 97.7% | 91.8% | 83.8% |
| Ant | 83.5% | 89.6% | 103.9% | 95.3% | 94.2% | 80.2% |
| Average | 93.8% | 96.4% | 99.7% | 96.4% | 94.6% | 83.6% |

**Buffer adjustment interval**    Table 11 shows the performance with different buffer adjustment intervals (denoted as $\Delta_b$) in different environments. We also include the special case of never adjusting the buffer capacity, i.e., $\Delta_b = \infty$. From Table 11, we find that the sensitivities of the buffer adjustment interval among different environments are similar. We find the performance is not sensitive to the buffer adjustment interval $\Delta_b$. We only observe an apparent performance degradation when the buffer adjustment interval is too large such that the policy distance cannot be reduced promptly.

Table 11: Sensitivity analysis on buffer adjustment interval.

| Environment | $\Delta_b =$ $1 \times 10^3$ | $\Delta_b =$ $3 \times 10^3$ | $\Delta_b =$ $1 \times 10^4$ | $\Delta_b =$ $3 \times 10^4$ | $\Delta_b =$ $1 \times 10^5$ | No adjustment $\Delta_b = \infty$ |
|---|---|---|---|---|---|---|
| HalfCheetah | 99.6% | 100.6% | 99.8% | 99.5% | 100.5% | 95.6% |
| Hopper | 96.7% | 94.8% | 97.0% | 93.8% | 94.8% | 94.5% |
| Walker2d | 96.6% | 103.1% | 98.1% | 101.2% | 96.1% | 90.2% |
| Ant | 97.5% | 92.4% | 103.9% | 103.3% | 79.7% | 80.0% |
| Average | 97.6% | 97.7% | 99.7% | 99.5% | 92.8% | 90.1% |

**Buffer policy distance threshold**    Table 12 shows the performance with different buffer policy distance thresholds (denoted as $D_0$) in different environments. We also include the special case of never adjusting the buffer capacity, i.e., $D_0 = \infty$. From Table 12, we find that the sensitivities of the buffer policy distance threshold among different environments are similar. It is shown that a very small threshold may reduce the performance, while the dynamic-capacity buffer helps improve the performance in a wide range of the threshold, especially when $D_0 = 0.1$ or $0.2$.

Table 12: Sensitivity analysis on buffer policy distance threshold.

| Environment | $D_0 = 0.05$ | $D_0 = 0.1$ | $D_0 = 0.2$ | $D_0 = 0.3$ | $D_0 = 0.5$ | No adjustment $D_0 = \infty$ |
|---|---|---|---|---|---|---|
| HalfCheetah | 90.8% | 102.9% | 99.8% | 98.8% | 99.2% | 95.6% |
| Hopper | 99.7% | 98.0% | 97.0% | 96.1% | 93.5% | 94.5% |
| Walker2d | 85.3% | 94.8% | 98.1% | 90.0% | 89.5% | 90.2% |
| Ant | 97.9% | 100.2% | 103.9% | 85.9% | 83.5% | 80.0% |
| Average | 93.1% | 99.0% | 99.7% | 92.7% | 91.4% | 90.1% |

**Multi-step delay**    Table 13 shows the performance with different multi-step delays (denoted as $T_0$) in different environments. We also include special cases of no delay, i.e., $T_0 = 0$, and delay all the time, i.e., $T_0 = \infty$. From Table 13, we find that the sensitivities of the multi-step delay among different environments are similar. Compared to the multi-step method without delay, the performance can be improved by using a delay mechanism in the early stage of training, also outperforming the one-step scheme. In addition, the performance is not very sensitive to the multi-step delay $T_0$, as we find that performance gains with different delays are similar.

Table 13: Sensitivity analysis on multi-step delay.

| Environment | No delay $T_0 = 0$ | $T_0 =$ $1 \times 10^5$ | $T_0 =$ $2 \times 10^5$ | $T_0 =$ $3 \times 10^5$ | $T_0 =$ $4 \times 10^5$ | $T_0 =$ $5 \times 10^5$ | one-step $T_0 = \infty$ |
|---|---|---|---|---|---|---|---|
| HalfCheetah | 87.1% | 95.7% | 99.2% | 99.8% | 100.8% | 95.5% | 96.5% |
| Hopper | 100.1% | 104.5% | 101.0% | 97.0% | 96.8% | 100.3% | 77.9% |
| Walker2d | 100.1% | 98.1% | 97.5% | 98.1% | 102.1% | 99.2% | 73.9% |
| Ant | 83.3% | 81.6% | 90.7% | 103.9% | 103.3% | 98.2% | 103.9% |
| Average | 92.7% | 95.0% | 97.1% | 99.7% | 100.1% | 98.3% | 88.1% |

## C.9 ADDITIONAL RESULTS IN HUMANOID-V3

In this subsection, we investigate the effect of RLx2 in Humanoid-v3, one of the control tasks from MuJoCo. Humanoid-v3 is considered relatively complex due to the high input dimensionality (376). Thus, apart from the standard 256 neurons in each hidden layer (same as other environments), we also train a dense model with 1024 neurons in each hidden layer. As shown in Table 14, the model with more hidden parameters (1024 hidden dimensions) does not achieve a better performance than a small model (256 hidden dimensions). This implies that the latter one seems to have sufficient capacity for the control task in Humanoid-v3.

Table 14: RLx2-TD3 in Humanoid-v3. We train two dense models with 256 and 1024 neurons in each hidden layer, respectively. We apply RLx2 and baseline algorithms in these two dense models. The table shows the average returns of each algorithm.

| Model | 256 hidden neurons | 1024 hidden neurons |
|---|---|---|
| Dense | $5721.8 \pm 173.0$ | $5097.3 \pm 228.9$ |
| Tiny | $3835.0 \pm 762.5$ | $3835.0 \pm 762.5$ |
| SS | $4248.4 \pm 1588.7$ | $4916.2 \pm 1058.5$ |
| SET | $5536.4 \pm 152.0$ | $5829.9 \pm 295.5$ |
| RigL | $5070.5 \pm 335.2$ | $5147.0 \pm 493.3$ |
| RLx2 | $5482.0 \pm 603.0$ | $5899.1 \pm 520.6$ |
| Actor Sparsity (%) | 92 | 99 |
| Critic Sparsity (%) | 85 | 98 |
| Number of Parameters | $\sim 63000$ | $\sim 72000$ |

In addition, Table 14 shows the performance of the sparse models trained with RLx2 and other baseline algorithms, where RLx2 succeeds in training a highly sparse model in Humanoid-v3 with performance degradation less than 5%. In particular, RLx2 can achieve sparsity of around 90% in a small model with only 256 hidden neurons, showing its robustness for complex control tasks. Also, RLx2 outperforms most of the baseline algorithms. Although SET shows comparable performance with RLx2, RLx2 shows much higher sample efficiency than SET according to Figure 11.

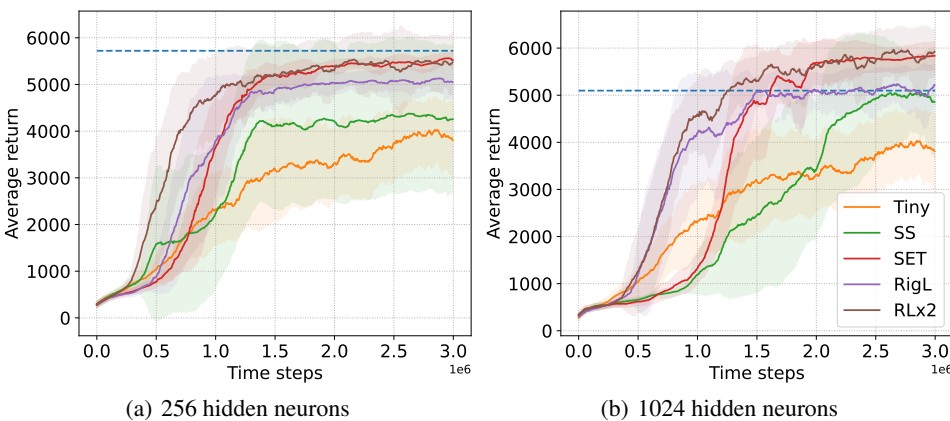

(a) 256 hidden neurons    (b) 1024 hidden neurons

Figure 11: Training processes of RLx2-TD3 in Humanoid-v3. The performance of each method is calculated as the average reward per episode over the last 30 evaluations of the training.

When applying RLx2 to larger models with 1024 neurons in each hidden layer, we find that it still performs well with extremely high sparsity (around 99%). We calculate the number of parameters of the two sparse models and find they are very close, as shown in Table 14. It suggests that RLx2 is an effective way to train a sparse model with the least parameters, and is robust to the hidden width of the dense model counterpart.

## C.10    Supplementary Results for Investigation of the Lottery Ticket Hypothesis

We provide additional experiments with TD3 for investigation of the lottery ticket hypothesis (LTH) in the other three environments, including HalfCheetah-v3, Hopper-v3, and Walker2d-v3. As shown in Figure 12, winning tickets fail to achieve the same performance as the dynamic topology in the reinforcement learning setting, while they all perform well under behavior cloning. This shows the necessity for reinforcement learning to adjust the network structure during the training process.

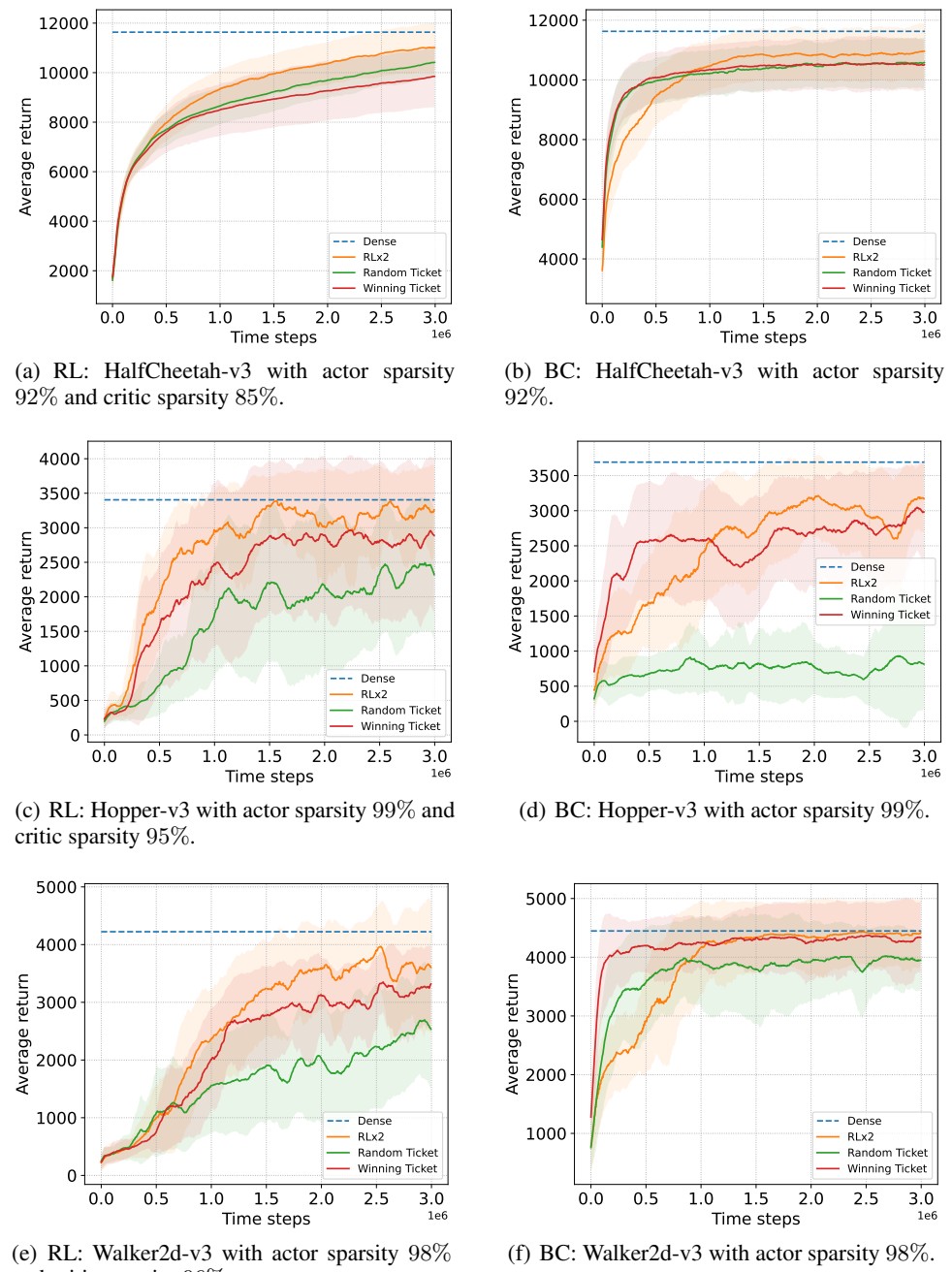

(a) RL: HalfCheetah-v3 with actor sparsity 92% and critic sparsity 85%.

(b) BC: HalfCheetah-v3 with actor sparsity 92%.

(c) RL: Hopper-v3 with actor sparsity 99% and critic sparsity 95%.

(d) BC: Hopper-v3 with actor sparsity 99%.

(e) RL: Walker2d-v3 with actor sparsity 98% and critic sparsity 96%.

(f) BC: Walker2d-v3 with actor sparsity 98%.

Figure 12: Comparisons of different methods for training a sparse DRL agent, where RLx2 is instantiated with TD3. Here, "RL" stands for the reinforcement learning setting, and "BC" stands for the behavior cloning setting.

## C.11 VISUALIZATION OF SPARSE MODELS

In this section, we show visualizations of the sparse networks obtained under RLx2 in our experiments. Note that each layer in the sparse network in our implementation is bound to a binary mask $M_{\theta_l}$, i.e., $\theta_l = \theta_l \odot M_{\theta_l}$ for the $l$-th layer. In the rest of this section, we investigate the property of the binary masks with visualization.

By visualizing the final mask after training, we find RLx2 drops redundant dimensions in raw inputs adaptively and efficiently. Specifically, Figure 13 shows the distribution of connections to each state dimension in Ant-v3. The figure shows that the resulting topology almost has no connections for input dimensions $28 - 111$, and the connections heavily concentrate on the first 27 dimensions. This observation indicates that the topology evolution method drops redundant dimensions in raw inputs adaptively and efficiently. A similar observation has also been made in Vischer et al. (2022), where it is shown that by iterative magnitude training, input dimensions irrelevant to the task are pruned entirely to help yield a minimal task-relevant representation.

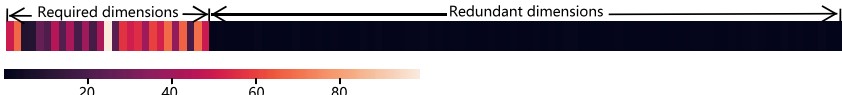

Figure 13: Distribution of connections to each state dimension in Ant-v3. Connections mainly concentrate on the first 27 dimensions.

In Figure 14, we further show visualizations of the raw binary masks (i.e., matrices with items representing neuron connections) of the actor in Ant-v3, where a black dot denotes an existing connection between the corresponding neurons, and a white point means there is no connection. As mentioned above, only very few connections are kept for the redundant input dimensions. We also find that the connections in the hidden layers tend to concentrate on a subset of neurons, showing that different neurons can play different roles in representations.

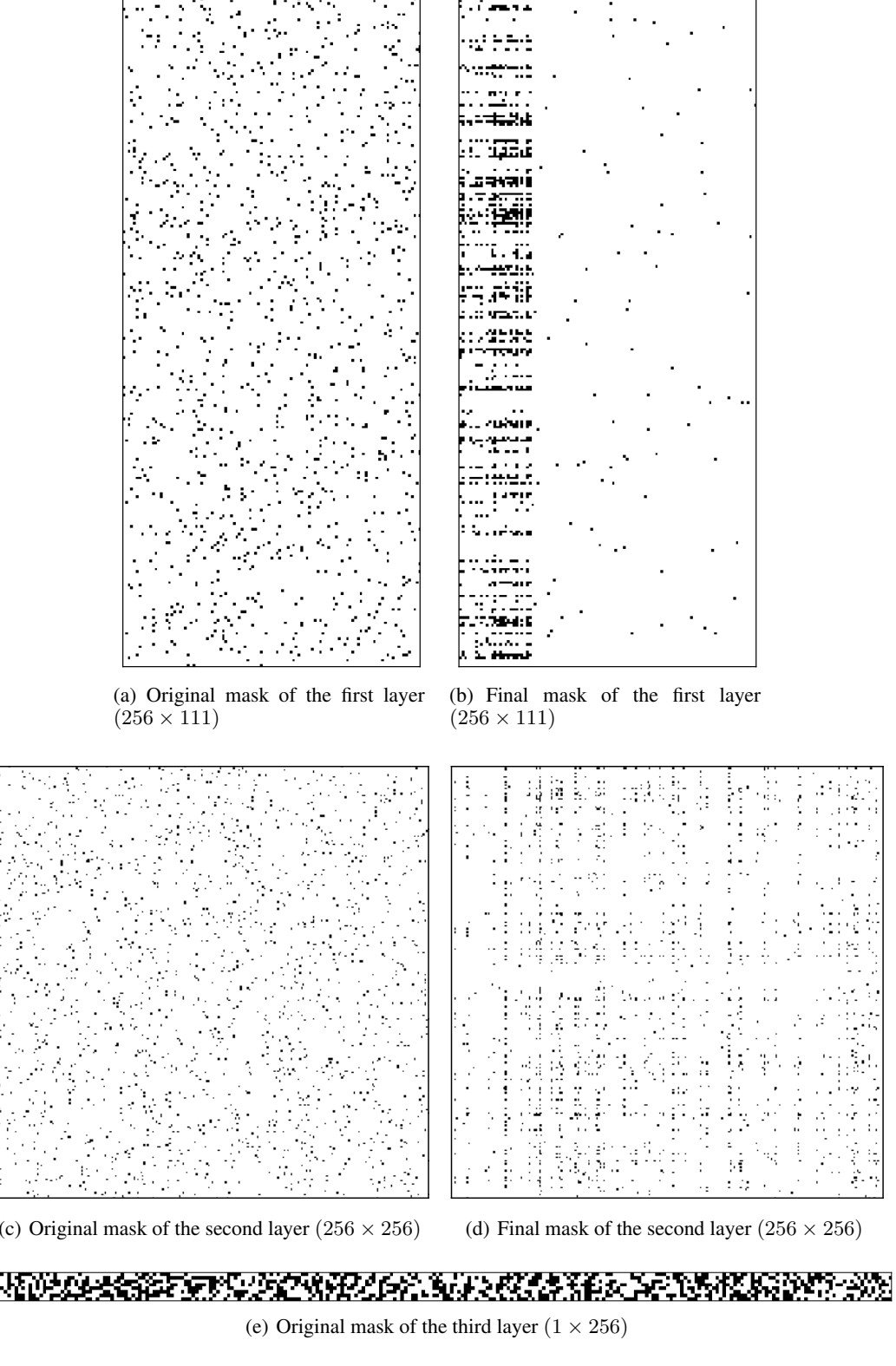

(a) Original mask of the first layer ($256 \times 111$)

(b) Final mask of the first layer ($256 \times 111$)

(c) Original mask of the second layer ($256 \times 256$)

(d) Final mask of the second layer ($256 \times 256$)

(e) Original mask of the third layer ($1 \times 256$)

(f) Final mask of the third layer ($1 \times 256$)

Figure 14: Visualization of the binary masks of the actor.

