# OpenReview forum: "RLx2: Training a Sparse Deep Reinforcement Learning Model from Scratch"
_ICLR.cc/2023/Conference — ICLR 2023 notable top 25%_

### Official Review · Reviewer_i62R · 2022-10-22

**Confidence:** 3
**Correctness:** 3
**Technical Novelty And Significance:** 2
**Empirical Novelty And Significance:** 2
**Recommendation:** 6

**Clarity, Quality, Novelty And Reproducibility:**

- The organization is clear and the technical approaches are well motivated.
- A little bit overclaim. The scope of the method in this paper should be in the off-policy TD learning area rather than the entire DRL community. For other DRL methods, such as the on-policy methods, it's unclear if the method is applicable or not.
- The novelty of method may be limited. Though I acknowledge the interesting empirical observations, the proposed technical solutions (gradient-based topology evolution, multi-step TD-targets, and thresholding the policy distance) are kind of trivial in practice.

**Strength And Weaknesses:**

Strengths:
- Training ultra-sparse DRL model from scratch is important while underexplored topic in practice.
- The contributions of this paper are clearly summarized, and the motivations are grounded in empirical observations.
- Experimental results show significantly better performance than the listed baselines.

Weaknesses:
- Since the one-step and multi-step TD-targets are used in a hybrid way in training, it's not a proper claim that a multi-step TD target could overcome the issue of inaccurate value estimation from a sparse network.
- Also, from the Eq. (4), it's unclear why the multi-step TD-targets could overcome the unreliable and inaccurate value estimations.
- The dynamic size of replay buffer is controlled by a pre-defined threshold on the policy distance, which is kind of heuristic in practice.
- Moreover, transitions are actually **randomly** selected from the replay buffer (to keep samples iid) rather than using the oldest top-K transitions. Therefore, the policy distance measure (Eq. (6)) may not effectively evaluate the inconsistency between behavior policy and current target policy.
- From Table 3, it's curious to me that the performance degrades when more than 3 steps are used. On the Hal. and Ant. benchmarks, the best number of steps is 2. Since it's a consistent observation that the performance would degrade with too many steps, this paper should present more in-depth discussion to explain this phenomenon.



**Summary Of The Paper:**

This paper presents the first work to train a sparse DRL agent from scratch. The major contributions are that the robust value estimation and effecient topology evolution are important to train a sparse DRL agent, and this paper proposes to leverage the gradient for network topology evolution and multi-step TD-target with dynamic buffer for robust value estimation. Experimental results are presented using two off-policy TD learning DRL models, and show the best performance over existing approaches.

**Summary Of The Review:**

Beyond the technical side, this paper is of good readability (well organized, empirically grounded motivations, etc). However, in terms of the technical novelty, the proposed solutions are a little bit trivial in sense.

---

> ### Author Response · Authors · 2022-11-15
> **Response to Reviewer i62R (part 1 of 2)**
>
> Thank you very much for your valuable feedback! In the following, we respond to your comments in detail.
> ### Weaknesses
> > Since the one-step and multi-step TD-targets are used in a hybrid way in training, it's not a proper claim that a multi-step TD target could overcome the issue of inaccurate value estimation from a sparse network.
>
> We have revised the statements in our revision, where the hybrid method of using one-step and multi-step TD-targets in training is now called the delayed multi-step mechanism, and it is the delayed multi-step TD target that overcomes the issue of inaccurate value estimation from a sparse network.
>
> > Also, from the Eq. (4), it's unclear why the multi-step TD-targets could overcome the unreliable and inaccurate value estimations.
>
> Multi-step TD targets have been shown to be effective in a number of DRL algorithms (e.g., [1], [2]) due to more efficient credit assignment, which provides a learning signal grounded by real data from the environment. This is important in sparse training since the networks are much sparser than the traditional dense networks, which can be prone to estimation error. Therefore, solely relying on single-step Q-value estimates from sparse neural networks can result in unreliable value estimates and a large bias, while multi-step Q-targets are grounded by real data. In addition, Eq. (4) also directly implies that introducing a multi-step target reduces the network fitting error by a multiplicative factor $\gamma^n$ for $n>1$.
>
> > The dynamic size of replay buffer is controlled by a pre-defined threshold on the policy distance, which is kind of heuristic in practice.
>
> In fact, The performance of our algorithm in different environments is insensitive to the policy distance such that we take the same value in different environments. In addtion, more evidence of the insensitivity of the policy distance can also be found in Appendix C.8 in our revision.
>
> > Moreover, transitions are actually randomly selected from the replay buffer (to keep samples iid) rather than using the oldest top-K transitions. Therefore, the policy distance measure (Eq. (6)) may not effectively evaluate the inconsistency between behavior policy and current target policy.
>
> Since the oldest samples may differ most from the current policy due to the rapid changing of the policy [4], the policy distance measure (Eq. (6)) provides an upper bound for the inconsistency between the behavior policy and current target policy in general. This policy distance measure is also regarded as the "age of experiences" in the replay buffer in prior works [3].
>
> > From Table 3, it's curious to me that the performance degrades when more than 3 steps are used. On the Hal. and Ant. benchmarks, the best number of steps is 2. Since it's a consistent observation that the performance would degrade with too many steps, this paper should present more in-depth discussion to explain this phenomenon.
>
> In the case when $n=1$, multi-step TD learning becomes the one-step TD learning (which can result in large bias) typically used in a number of DRL algorithms. When $n$ is the trajectory length, multi-step TD is the Monte-Carlo method, which may incur high variances. Therefore, an appropriate $n$ can provide a good trade-off between bias and variance. In fact, this phenomenon has been observed in [5] on multi-step TD targets, showing that a smaller value of $n$ is often the most effective. Recent works applying multi-step TD targets, e.g., [1], [2], also use small $n$ values, i.e., 3 or 5, respectively.

---

> > ### Author Response · Authors · 2022-11-17
> > **Response to Reviewer i62R (part 2 of 2)**
> >
> > ### Clarity, Quality, Novelty And Reproducibility
> >
> > > A little bit overclaim. The scope of the method in this paper should be in the off-policy TD learning area rather than the entire DRL community. For other DRL methods, such as the on-policy methods, it's unclear if the method is applicable or not.
> >
> > Our RLx2 framework is a general algorithm framework for off-policy TD learning algorithms, which is applicable to a wide range of off-policy algorithms. Besides, TD3 [6] and SAC [7] are two state-of-the-art off-policy algorithms and are likely to outperform on-policy algorithms, such as PPO, in many scenarios. In our experiments, we evaluated our RLx2 framework with SOTA algorithms TD3 and SAC in sparse models, as in prior works, e.g., [8]. Our empirical results showed the superior performances of RLx2 for TD3 and SAC. For other DRL methods, such as the on-policy methods, it will be interesting future work to extend RLx2 to these areas.
> >
> > > The novelty of method may be limited. Though I acknowledge the interesting empirical observations, the proposed technical solutions (gradient-based topology evolution, multi-step TD-targets, and thresholding the policy distance) are kind of trivial in practice.
> >
> > 1. In this paper, we focus on designing a sparse training framework to enable sparse DRL training from scratch, which owns a large potential to significantly reduce computation expenditure, enable efficient deployment on resource-limited devices, and achieve excellent flexibility in model adaptation. We investigated the fundamental obstacles (as shown in Fig. 2 in the paper) in training a sparse DRL agent from scratch. Our first major novelty is discovering two critical factors for achieving good performance under sparse training in DRL, robust value estimation and efficient topology exploration.
> >
> > 2. Our second major novelty is the development of RLx2, which is motivated by our key findings. RLx2 is not a trivial integration of the proposed technical solutions into the RL task. Instead, RLx2 possesses two key functions, i.e., efficient topology exploration and robust value learning. In particular, our proposed technical solutions (gradient-based topology evolution, multi-step TD-targets, and thresholding the policy distance) are simple yet effective in simultaneously enabling these two key functions. Our empirical results show the state-of-the-art performance of RLx2 compared to other representative benchmarks, i.e., achieving up to $20\times$ acceleration in training and $50\times$ in inference in FLOPs.
> >
> >
> > ***
> >
> > ***Finally, we thank the reviewer again for the valuable comments. If our response resolves your concerns satisfactorily, we want to kindly ask the reviewer to consider raising the score rating of our work. We will also be happy to answer any further questions you may have during the discussion.***
> >
> > ***
> >
> >
> > ```
> > [1] Hessel, Matteo, Joseph Modayil, Hado Van Hasselt, Tom Schaul, Georg Ostrovski, Will Dabney, Dan Horgan, Bilal Piot, Mohammad Azar, and David Silver. "Rainbow: Combining improvements in deep reinforcement learning." In Thirty-second AAAI conference on artificial intelligence. 2018.
> > [2] Kozuno, Tadashi, Yunhao Tang, Mark Rowland, Rémi Munos, Steven Kapturowski, Will Dabney, Michal Valko, and David Abel. "Revisiting Peng’s Q ($ λ $) for Modern Reinforcement Learning." In International Conference on Machine Learning, pp. 5794-5804. PMLR, 2021.
> > [3] Fedus, W., Ramachandran, P., Agarwal, R., Bengio, Y., Larochelle, H., Rowland, M., & Dabney, W. (2020, November). Revisiting fundamentals of experience replay. In International Conference on Machine Learning (pp. 3061-3071). PMLR.
> > [4] Fujimoto, S., Hoof, H., & Meger, D. (2018, July). Addressing function approximation error in actor-critic methods. In International conference on machine learning (pp. 1587-1596). PMLR.
> > [5] Sutton, R. S., & Barto, A. G. (2018). Reinforcement learning: An introduction. MIT press.
> > [6] Fujimoto, Scott, Herke Hoof, and David Meger. "Addressing function approximation error in actor-critic methods." In International conference on machine learning, pp. 1587-1596. PMLR, 2018.
> > [7] Haarnoja, Tuomas, Aurick Zhou, Pieter Abbeel, and Sergey Levine. "Soft actor-critic: Off-policy maximum entropy deep reinforcement learning with a stochastic actor." In International conference on machine learning, pp. 1861-1870. PMLR, 2018.
> > [8] Björck, Johan, Xiangyu Chen, Christopher De Sa, Carla P. Gomes, and Kilian Weinberger. "Low-precision reinforcement learning: Running soft actor-critic in half precision." In International Conference on Machine Learning, pp. 980-991. PMLR, 2021.
> > ```

---

> ### Author Response · Authors · 2022-11-22
> **Reminder to Reviewer i62R**
>
> Dear reviewer,
>
> Thank you for your effort in reviewing our paper!
>
> We wonder whether our reply fully addresses your concerns. If so, could you please consider raising your score for our work?
> Please let us know if you have any further questions. We will be more than happy to discuss this with you and answer any remaining questions.
>
> Thank you very much!

---

### Official Review · Reviewer_7aAy · 2022-10-25

**Confidence:** 3
**Correctness:** 3
**Technical Novelty And Significance:** 3
**Empirical Novelty And Significance:** Not applicable
**Recommendation:** 8

**Clarity, Quality, Novelty And Reproducibility:**

The whole paper is written clearly. The work is with good reproducibility with a detailed introduction of the details. Considering that the core method is RigL, the novelty is limited at some level. But the studies on the RL are in detail.


**Strength And Weaknesses:**

Strength
- The proposed method is carefully designed and well implemented. It can maintain good performance with very high-level sparsity on different datasets. The proposed method integrates the topology evolution method into the RL task very well.
- The details of the algorithms and experiments are discussed and analyzed carefully, making the work easy to reproduce.
- The paper is easy to follow.

Weakness
- The proposed method is an integration of the RigL method into the RL task and framework. The major contribution is on how to integrate it into the RL framework. This limits the significance of the proposed method, especially the lottery ticket based method for sparse network learning has been studied wildly.
- Apart from RigL, the authors may consider studying how different sparse network learning methods can help RL in what kind of way. This may generalize the proposed techniques to a more general sparse network based RL framework.


**Summary Of The Paper:**

This paper proposes to learn an efficient sparse network for reinforcement learning (RL). It learns the sparse network from scratch instead of relying on knowledge distillation or pruning. The proposed method is designed based on the gradient-based topology evolution criteria. Experiments show that the proposed method can learn very sparse networks with satisfying performance.

**Summary Of The Review:**

The major contribution is about how to integrate the RigL based sparse network learning approach into the RL framework, which limits the novelty. But the work conducts very detailed design and studies and analyses of the sparse network learning for RL. Many learning strategies are investigated.

The rebuttal addressed my concerns. I am happy for this paper to be accepted.

---

> ### Author Response · Authors · 2022-11-15
> **Response to Reviewer 7aAy**
>
> Thank you very much for your valuable feedback. In the following, we respond to your comments in detail.
>
> > The proposed method is an integration of the RigL method into the RL task and framework. The major contribution is on how to integrate it into the RL framework. This limits the significance of the proposed method, especially the lottery ticket based method for sparse network learning has been studied wildly.
>
>  1)  Our first main contribution is to investigate the fundamental obstacles (as shown in Fig. 2 in the paper) in training a sparse DRL agent from scratch. We discover two critical factors for achieving good performance under sparse training, namely robust value estimation and efficient topology exploration.
>
>  2)  Our second main contribution is the development of RLx2, which is motivated by our key findings. RLx2 is not a trivial integration of the RigL method into the RL task. Instead, RLx2 possesses two key functions, i.e., a gradient-based search scheme for efficient topology exploration and a delayed multi-step TD target mechanism with a dynamic-capacity replay buffer for robust value learning. It also achieves up to $20\times$ acceleration in training and $50\times$ in inference in FLOPs. In fact, a direct integration of the SET (a comparable topology evolution schema as RigL) method into the RL task has been proposed by [Sokar et al. (2021)], which only brings about 50% sparsity. The RigL method has also been integrated into the RL task in [Graesser et al. (2022)], whose performance is also below our RLx2 algorithm.
>
>  3)  Although the lottery ticket-based method for sparse network learning has been studied widely, sparse training from scratch, especially for the DRL setting, is far less studied. We are among the first to find that robust value estimation and efficient topology exploration are critical in archiving ultra-sparse DRL training from scratch. We also compared our method with representative dynamic sparse training benchmarks, including SET and RigL in our experiments, showing the superiority of our proposed RLx2 algorithm, i.e., achieving up to $20\times$ acceleration in training and $50\times$ in inference in FLOPs.
>
> > Apart from RigL, the authors may consider studying how different sparse network learning methods can help RL in what kind of way. This may generalize the proposed techniques to a more general sparse network based RL framework.
>
> In this paper, we focus on the sparse training from scratch in DRL. We have compared representative dynamic sparse training benchmarks, including SET and RigL, and other sparse training schemas, e.g., static sparse, in our experiments and show that RLx2 outperforms all other benchmarks. The reviewer's suggestion for further investigating this generalization is very interesting, and we plan to study it in our future work.
> ***
> ***Finally, we thank the reviewer again for the valuable comments. If our response resolves your concerns satisfactorily, we would like to kindly ask the reviewer to consider raising the score rating for our work. We will also be happy to answer any further questions you may have during the discussion.***

---

> ### Author Response · Authors · 2022-11-22
> **Reminder to Reviewer 7aAy**
>
> Dear reviewer,
>
> Thank you for your effort in reviewing our paper!
>
> We wonder whether our reply fully addresses your concerns. If so, could you please consider raising your score for our work?
> Please let us know if you have any further questions. We will be more than happy to discuss this with you and answer any remaining questions.
>
> Thank you very much!

---

### Official Review · Reviewer_Qrdc · 2022-10-25

**Confidence:** 4
**Correctness:** 4
**Technical Novelty And Significance:** 3
**Empirical Novelty And Significance:** 3
**Recommendation:** 8

**Clarity, Quality, Novelty And Reproducibility:**

The paper is clear, well-written, and makes novel scientific contributions. Pseudocodes of algorithms and their hyper-parameters are provided in the appendix.

**Strength And Weaknesses:**

The RigL method learns a sparse topology from scratch. Unlike other model compression methods, there is no need for pre-training a dense network, which markedly reduces the computational requirements of both training and inference. RLx2 could therefore greatly benefit online learning problems on resource-constrained devices. I appreciate that the author has validated the importance of robust value estimation and its effect on the topology learning of RigL.

There are two main weaknesses in the paper. The first is the lack of proper sensitivity analysis for some of the new algorithm hyper-parameters. Although the hyper-parameters are the same for all four environments, it is unclear how sensitive the performance is to each hyper-parameter and how the sensitivity differs between environments. These hyper-parameters include the initial mask update fraction, mask update interval, buffer adjustment interval, buffer policy distance threshold, and multi-step delay.

The second weakness relates to the computational complexity of the algorithm. The topology learning of RigL consumes O(N logN) time with N as the number of weights in the corresponding dense network. Although the topology is updated every 10000 steps and the cost may be negligible compared to the main RL algorithm in these experiments, 10000 is still a constant. With the exponentially increasing number of parameters, in the future the computational requirement of the topology learning step might begin to dominate and grow faster than the main algorithm, making it less scalable. It would be great if the dropping and growing of connections could be done locally for each connection instead of requiring a global sort. This issue concerns the RigL method adopted by this paper and not the techniques introduced in this paper for making the value estimate robust.

Here are some more questions and comments:
* What do the shaded areas in the plots represent?
* Did you experiment with using multi-step TD targets from the beginning? What did you observe?
* What happens if you do not anneal the update fraction, but keep it constant at a smaller value?
* Should the two robustification techniques be applied to the original algorithms even when not using RigL?
* “We also show in Appendix C.2 that the performance of RLx2 is insensitive to the policy threshold.” I do not see any sensitivity plots in C.2.
* “Upon any update to \phi and appending new transitions to…” Is this sentence correct? Was the buffer adjustment not done every \Delta_b steps?

**Summary Of The Paper:**

Sparse neural networks that have an order of magnitude fewer parameters than a dense network without a significant drop in performance can greatly reduce computational requirements of reinforcement learning (RL) agents, allowing their deployment on resource-limited devices. Different sparse network topologies can have drastic differences in performance. A method for the training of sparse networks from scratch, known as RigL, learns the topology throughout training by dropping low-magnitude weights and growing connections that have large gradients. This paper extends RigL to the deep RL domain by adding two techniques for making the value function estimation more robust. One technique is using multi-step temporal difference targets instead of one-step ones. The other is dynamically changing the size of the replay buffer if its stored actions get too far from the current policy. The paper demonstrates with examples that robust value learning is essential for improving the performance of RigL since, in part, the value function affects the gradient-based topology learning of RigL. Experiments with RLx2 using SAC and TD3 algorithms on four mujoco environments show significant model compression with minimal loss in performance.

**Summary Of The Review:**

The paper establishes that having a robust value estimator is important when learning the sparse network topology using RigL in Deep RL. The method trains a sparse network from scratch and achieves significant model compression with minimal loss in performance. Claims are backed up with experiments. The two introduced techniques are simple yet effective for applying RigL to Deep RL. The paper can be improved by adding sensitivity analyses for the new hyper-parameters as discussed previously.

---

> ### Author Response · Authors · 2022-11-15
> **Response to Reviewer Qrdc (Part 1 of 2)**
>
> Thank you very much for your positive and valuable feedback! In the following, we respond to your comments in detail.
>
> ### Weaknesses
>
> > The first is the lack of proper sensitivity analysis for some of the new algorithm hyper-parameters. Although the hyper-parameters are the same for all four environments, it is unclear how sensitive the performance is to each hyper-parameter and how the sensitivity differs between environments. These hyper-parameters include the initial mask update fraction, mask update interval, buffer adjustment interval, buffer policy distance threshold, and multi-step delay.
>
> We provide more empirical validations to study the sensitivity of the performance to each hyper-parameter mentioned above and to investigate how the sensitivity level differs in different environments. The new results are shown in Appendix C.8 in our revision.
>
> > The second weakness relates to the computational complexity of the algorithm. The topology learning of RigL consumes O(N logN) time with N as the number of weights in the corresponding dense network. Although the topology is updated every 10000 steps and the cost may be negligible compared to the main RL algorithm in these experiments, 10000 is still a constant. With the exponentially increasing number of parameters, in the future the computational requirement of the topology learning step might begin to dominate and grow faster than the main algorithm, making it less scalable. It would be great if the dropping and growing of connections could be done locally for each connection instead of requiring a global sort. This issue concerns the RigL method adopted by this paper and not the techniques introduced in this paper for making the value estimate robust.
>
> In our design of RLx2, we actually took the computational overhead from topology evolution, including link dropping and growing, into account.
> - Link Dropping: It only requires sorting the remaining weights in the sparse models. Thus, the time complexity of the dropping operation is $O((1-s)N \log N)$, where $s$ is the sparsity level, which has been illustrated in the “Link Dropping” paragraph in Appendix A.1.
> - Link Growing: Instead of naively maintaining the global gradients of all parameters in dense models, we provided an efficient link growing algorithm in Alg. 2 (Appendix A.1), which improved the time complexity of link growing from $O(N\log N)$ to $O((1-s)N\log N)$. Thus, RLx2 is scalable. The main idea of Alg.2 is to utilize a data structure "heap" to speed up the top-K operation.
>
> It is worth emphasizing that our improvement also works for RigL. We agree with the reviewer that it will be interesting to enable the dropping and growing of links to be done locally for each link instead of requiring a global operation. We plan to investigate this interesting aspect in our future work.

---

> > ### Author Response · Authors · 2022-11-15
> > **Response to Reviewer Qrdc (Part 2 of 2)**
> >
> > ### Questions and comments
> >
> > > What do the shaded areas in the plots represent?
> >
> > The shaded areas refer to the standard deviation.
> >
> > > Did you experiment with using multi-step TD targets from the beginning? What did you observe?
> >
> > We conducted an experiment on multi-step TD targets from the beginning, with results given in Table 13 in the Appendix. We observed that using multi-step TD targets from the beginning will reduce the performance compared to our delayed scheme. This is because the policy changes rapidly in the beginning. Thus, at the early stage of the training, there is a large gap between the policy that generates the training transitions and the current learning policy, resulting in a large error between the multi-step TD targets and true state values.
> >
> > > What happens if you do not anneal the update fraction, but keep it constant at a smaller value?
> >
> > We provide a supplemental experiment on using small constant update fraction $\zeta$ instead of annealing during the topology evolution process, with results given in the following table. We find that our schema of annealing the update fraction achieves higher performance.
> >
> > |Scheme | HalfCheetah | Hopper | Walker2d | Ant | Avearge|
> > |----|----|----|----|----|----|
> > |Fixed $\zeta=0.1$ | 99.5 | 96.3 | 96.5 | 66.9 | 89.8|
> > |Fixed $\zeta=0.3$ | 94.5 | 95.8 | 94.3 | 85.5 | 92.0|
> > |Fixed $\zeta=0.5$ | 95.2 | 85.9 | 93 | 83.5 | 89.4|
> > |Annealing (Ours) | 99.8 | 97.0 | 98.1 | 103.9 | 99.7|
> >
> > > Should the two robustification techniques be applied to the original algorithms even when not using RigL?
> >
> > TD3 and SAC are two commonly accepted SOTA off-policy RL algorithms. The main contribution of our paper is designing a framework that trains a sparse DRL agent from scratch with comparable performance to SOTA algorithms such as TD3 and SAC. Existing sparse training works also compare their algorithm with these baseline algorithms, e.g., Lee et al., 2021, Sokar et al., 2021 and Graesser et al., 2022. It will be interesting future work to apply our robustification techniques to these baseline algorithms.
> >
> > > “We also show in Appendix C.2 that the performance of RLx2 is insensitive to the policy threshold.” I do not see any sensitivity plots in C.2.
> >
> > We have revised the paper, and the experimental results on the sensitivity of the policy threshold can be found in Appendix C.8 in our revision.
> >
> > >  “Upon any update to \phi and appending new transitions to…” Is this sentence correct? Was the buffer adjustment not done every \Delta_b steps?
> >
> > We have revised it in our revision.
> > ***
> > Finally, we want to thank the reviewer again for the detailed and valuable comments. Please let us know if you have any further questions, and we will be happy to answer them.

---

### Official Review · Reviewer_B2aD · 2022-10-30

**Confidence:** 4
**Correctness:** 3
**Technical Novelty And Significance:** 3
**Empirical Novelty And Significance:** 3
**Recommendation:** 8

**Clarity, Quality, Novelty And Reproducibility:**

Clarity: The paper is clear and reads well, in my opinion.

Quality: The paper has a good quality level, and it seems to be well executed.

Originality: Good level of novelty as reflected also by the proper related work discussion.

Reproducibility: I believe that it may be possible (but not straightforward) for a reader with a good level of experience on the topic to reproduce the proposed methodology given just the paper details, if enough time is put into it. Anyway, the authors promise the release of open-source code if accepted (but it is not presented in the supplementary material) and this shall increase seriously the reproducibility levels.


**Strength And Weaknesses:**

Strength:
* Timely, relevant, and under-studied topic, particularly for deep reinforcement learning. If enough efforts are put into it, and if it will become more mature, it has the potential of seriously decreasing the costs associated with deep reinforcement learning. Counterintuitively, as shown in the paper, it also has the potential of improving the state-of-the-art performance in deep reinforcement learning.
* The proposed method has a good level of novelty.
* The empirical validation is well-designed. The results show that the proposed method can outperform the baselines.
* Well written paper

Weaknesses – in my opinion, the paper doesn’t seem to have major flaws. Further, I would raise some points for discussion which are intriguing me:
* Why the Humanoid environment hasn’t been considered in the experiments? I assume that at this very high sparsity levels, given its large state space, the performance may be far from optimal. Can you consider adding an extra small experiment on Humanoid where you can study two network architectures (one like in the paper and one with more hidden neurons to compensate for the high sparsity regime) for RLx2 and the typical baselines used in the paper?
* From Table 2, I see that the proposed method (RLx2) consistently improves the dense SAC baseline in terms of performance even when over 90% sparsity is considered. At the same time, on TD3 the gain induced by RLx2 is considerably smaller. Do you have any idea why RLx2 impacts SAC in a different manner than TD3?

Other comments:
* In table 1, I believe that it is worth also adding the work of Graesser et al., 2022 for a more complete overview.
* page 4, I believe that “…links with the smallest weights.” is not accurate for negative weights. According with Alg 1, line 9, it is rather about … links with the smallest absolute value of the weights…
* page 5, “…Here we adopt the methods in (Frankle & Carbin, 2019) for…”. The following procedure is not exactly the one from the mentioned paper. I believe that it would be informative to clarify exactly what was taken from that paper and what was added by you.
* page 6, Dynamic capacity buffer – According with the description, the buffer size (capacity) is always increasing? Is my understanding correct?
* page 7, “…SET (Bellec et al., 2017),…” is not correct. SET was proposed by Mocanu et al., July 2017, if referring to the first arxiv version, or 2018 if referring to the year of formal publication.
* page 8, “…which replace the topology evolution scheme in RLx2 with Tiny, SS and SET, while keeping…” may be ok, but it is not sufficiently clear to me what are the changes as some of those methods are static and some are dynamic (evolves). Perhaps you can clarify it better in the paper.
* page 9 – Conclusion – a fair paper, with a good level of self-criticism, discusses also a bit of the limitations of the proposed methodology, while possibly indicating future work directions. These are missing in this paper.
 * page 25 – Appendix C.5 – Figures 9 ad 10 – it may be useful for the reader to see also somehow the sparsity levels in those plots without the need of scrolling back to Section 5.1



**Summary Of The Paper:**

The paper studies sparse training with dynamic sparsity in the context of extremely sparse neural network models, trained from scratch, as function approximators for deep reinforcement learning. Consequently, the paper proposes a new sparse training method for deep reinforcement learning, named the Rigged Reinforcement Learning Lottery (RLx2). According with the experimental evaluation on four continuous control task environments, RLx2 seems to perform considerably better than the sparse baseline methods considered. Moreover, it can also outperform in a handful number of cases the dense equivalent.

**Summary Of The Review:**

Overall, I believe that the paper has a good level of novelty, with good results, and a good presentation. I am curious to see the authors’ opinion about the comments raised above.

---

> ### Author Response · Authors · 2022-11-15
> **Response to Reviewer B2aD (Part 1 of 2)**
>
> Thank you very much for your positive and valuable feedback! In the following, we respond to your comments in detail.
>
> ### Weaknesses
> > Why the Humanoid environment hasn’t been considered in the experiments? I assume that at this very high sparsity levels, given its large state space, the performance may be far from optimal. Can you consider adding an extra small experiment on Humanoid where you can study two network architectures (one like in the paper and one with more hidden neurons to compensate for the high sparsity regime) for RLx2 and the typical baselines used in the paper?
>
> Thanks for your suggestion. In addition to the Ant task (already evaluated in our paper, whose state dimension is 111, considerably larger than other tasks), we follow the reviewer's suggestion and evaluate RLx2 on the higher dimensional Humanoid environment. From our results, we see that when training a model with 256 hidden neurons, RLx2 achieves a sparsity of 90% without much performance degradation. When applied to a larger model with 1024 hidden neurons, RLx2 can achieve an even higher sparsity of 99%. We provide more details about Humanoid experiments in Appendix C.9 in our revision.
>
> >  From Table 2, I see that the proposed method (RLx2) consistently improves the dense SAC baseline in terms of performance even when over 90% sparsity is considered. At the same time, on TD3 the gain induced by RLx2 is considerably smaller. Do you have any idea why RLx2 impacts SAC in a different manner than TD3?
>
> Please note that the performances of TD3 and SAC are evaluated under different sparsity levels. Specifically, the sparsity reported in Table 2 is roughly the maximum achievable sparsity without performance degradation (compared to the dense network counterpart). It can be found in Table 2 that the average sparsity used for SAC is slightly lower than that of TD3.
> We include the results for training RLx2-TD3 with the same sparsity as that for RLx2-SAC. As shown in the table below, we do not find a significant difference in performance between RLx2-TD3 and RLx2-SAC under the same sparsity.
> || HalfCheetah | Hopper | Walker2d | Ant | Average |
> |----|----|----|----|----|----|
> |Actor Sparsity| 90% | 98% | 90% | 90% | 92% |
> |Critic Sparsity| 80% | 95% | 90% | 75% | 85% |
> |RLx2-TD3 Performance| 101.6% | 97.0% | 111.3% | 109.7% | 104.9% |
> |RLx2-SAC Performance| 102.2% | 109.7% | 103.2% | 105.6% | 105.2%|

---

> > ### Author Response · Authors · 2022-11-15
> > **Response to Reviewer B2aD (Part 2 of 2)**
> >
> > ### Comments
> > > In table 1, I believe that it is worth also adding the work of Graesser et al., 2022 for a more complete overview.
> >
> > Thanks for pointing to the reference. We included the work of Graesser et al., 2022 for the topology evolution paradigm in Table 1 in our revision.
> >
> > > page 4, I believe that “…links with the smallest weights.” is not accurate for negative weights. According with Alg 1, line 9, it is rather about … links with the smallest absolute value of the weights…
> >
> > We have revised the sentence in our revision.
> >
> > > page 5, “…Here we adopt the methods in (Frankle & Carbin, 2019) for…”. The following procedure is not exactly the one from the mentioned paper. I believe that it would be informative to clarify exactly what was taken from that paper and what was added by you.
> >
> > Thanks for pointing this out. We use the same procedure for evaluating the sparse mask as in Frankle & Carbin, 2019, i.e., training with the static mask from scratch. However, we take a different way of generating the static sparse mask compared to Frankle & Carbin, 2019. Specifically, Frankle & Carbin, 2019 obtains the sparse mask, i.e., the "lottery ticket", by pruning a pretrained dense model since they focus on the pruning method. In our case, since we focus on dynamic sparse training, we evaluate the final-obtained masks of the dynamic sparse networks after training. We have clarified it better in our revision.
> >
> > > page 6, Dynamic capacity buffer – According with the description, the buffer size (capacity) is always increasing? Is my understanding correct?
> >
> > The dynamic capacity buffer keeps checking the policy distance and will drop the oldest transitions when their corresponding behavior policy is too far from the current policy. Also, there is a hard capacity upper bound ($B_{\text{max}}$) as detailed in Algorithm 3 in Appendix A.3.
> >
> > > page 7, “…SET (Bellec et al., 2017),…” is not correct. SET was proposed by Mocanu et al., July 2017, if referring to the first arxiv version, or 2018 if referring to the year of formal publication.
> >
> > Thanks for pointing this out. We have revised the reference in our revision.
> >
> > > page 8, “…which replace the topology evolution scheme in RLx2 with Tiny, SS and SET, while keeping…” may be ok, but it is not sufficiently clear to me what are the changes as some of those methods are static and some are dynamic (evolves). Perhaps you can clarify it better in the paper.
> >
> > Take RLx2 with TD3 (Algorithm 4 in Appendix B) as an example. Topology evolution takes place in Lines 16 and 21 of "Topology Evolution($\cdot$)", respectively. In our ablation study, we replace these two lines with our desired methods for topology evolution. Specifically, SET refers to evolving the topology by randomly growing the network links and dropping links with the smallest absolute weights periodically, and RLx2 (topology evolution criteria by RigL) refers to evolving topology by growing links with maximum gradients and dropping links with the smallest absolute weight periodically. In particular, Tiny (using static dense networks with small hidden dimensions) and SS (using static sparse networks) are also regarded as special topology evolution criteria, i.e., keeping topology static throughout. We have clarified this better in our revision.
> >
> > > page 9 – Conclusion – a fair paper, with a good level of self-criticism, discusses also a bit of the limitations of the proposed methodology, while possibly indicating future work directions. These are missing in this paper.
> >
> > We have discussed limitations and future work in our revision. Specifically,  two main limitations of our RLx2 framework includes: 1) only applicable to off-policy RL algorithms; 2) not applicable to multi-agent or offline RL settings.
> >
> > > page 25 – Appendix C.5 – Figures 9 ad 10 – it may be useful for the reader to see also somehow the sparsity levels in those plots without the need of scrolling back to Section 5.1
> >
> > We have annotated the sparsity levels in Figures 9 ad 10 in our revision.
> > ***
> > Finally, we want to thank the reviewer again for the detailed and valuable comments. Please let us know if you have any further questions, and we will be happy to answer them.

---

> > > ### Comment · Reviewer_B2aD · 2022-12-07
> > > **Nice answers - Thank you**
> > >
> > > I thank the authors for adressing my comments and for studying also the Humanoid environment. I believe that the paper is an excellent read overall, and I am keeping with a very high confidence my original 'accept, good paper'  recommendation.

---

> > > > ### Author Response · Authors · 2022-12-08
> > > > **Thanks**
> > > >
> > > > Thank you very much for your help in improving our paper! We really appreciate your positive and valuable response.

---

### Author Response · Authors · 2022-11-18
**Revision**

We thank all reviewers for the insightful comments and helpful suggestions. We have revised the paper accordingly, with revised parts marked in blue.

- We provide a comprehensive sensitivity analysis in Appendix C.8 for additional algorithm hyper-parameters, including the initial mask update fraction, mask update interval, buffer adjustment interval, buffer policy distance threshold, and multi-step delay.
- We include an experiment on a more complex environment with larger state space, Humanoid-v3, in Appendix C.9 to further evaluate RLx2 on complex environments, which achieves consistent performance improvement.
- We highlight our hybrid way of using one-step and multi-step TD targets as the "delayed multi-step mechanism".
- We include the work from (Graesser et al., 2022) in Table 1 for a more comprehensive comparison with prior works.
- We clarify the difference between our method and that of Frankle & Carbin, 2019, about the way to evaluate the sparse mask in Section 4.2.
- We discuss some limitations/future works in our conclusion section to reach a good level of self-criticism.

---

### Decision · Program_Chairs · 2023-01-20

**Decision:**

Accept: notable-top-25%

**Justification For Why Not Higher Score:**

The work benefits substantially due to a prior method, RigL, making the work an obvious win but not among the most excellent ones.


**Justification For Why Not Lower Score:**

The work is an important contribution to learning sparse networks for RL agents.

**Metareview: Summary, Strengths And Weaknesses:**

This work addresses the important problem of learning a reinforcement learning (RL) agent from scratch with a sparse network throughout. Recent development in sparse connection learning, for example RigL, allowed the authors to aim for this ambition. The authors proposed network-topology evolution through RigL, but recognized that robust value learning is important for good performance under sparse networks, which they enabled through a multi-step TD target and a dynamic-capacity replay buffer. The authors show substantial model compression and acceleration of training and inference with only a slight performance loss. The work successfully and clearly makes a significant contribution to an important problem; likewise, I recommend accepting the paper. I suggest mentioning the super-linear computational complexity of the topology evolution in the paper.



**Note From Pc:**

if the above contains the word "oral" or "spotlight" please see: "oral" presentation means -> notable-top-5% and "spotlight" means -> notable-top-25%. As stated in our emails, we are disassociating presentation type from AC recommendations